# MV-CLAM: Multi-View Molecular Interpretation with Cross-Modal Projection via Language Model

## Abstract

Large language models (LLMs) have shown significant potential in the biomolecular domain, particularly by demonstrating that effective adaptation of molecular representations for LLMs can greatly improve the quality of molecular captions. Most previous works have focused on aligning unimodal molecular structures with text, overlooking the diversity of modalities. Naive approaches to aligning multi-modal molecular structures with text often lead to (1) separately aligned embeddings, (2) inconsistent textual representations, and (3) increased computational overhead. To address these challenges, we propose LLM framework MV-CLAM equipped with MQ-Former, a novel multi-querying transformer. This architecture introduces a cross-model projector facilitating the simultaneous alignment of 2D and 3D molecular representations to a unified text token. By employing a shared self-attention layer, MQ-Former preserves rich molecular embeddings across different dimensions while consolidating them into a universal molecular token. Our approach outperforms baseline models in both molecule-text retrieval and molecule captioning tasks. Additionally, our framework shows promising results for zero-shot molecule editing and molecule-related question answering. By effectively integrating multi-view molecular data into a format conducive to LLMs, our method serves as a valuable tool for enhancing the characterization and understanding of chemical structures, facilitating a more seamless transition from molecular data to textual descriptions. The source code of MV-CLAM is available in `https://anonymous.4open.science/r/mv-clam-4827`.

## 1 Introduction

Given that human expertise relies on a deep understanding of molecular structures and biomedical text, advancing language models to effectively integrate the two domains is a logical step forward (Edwards et al., 2022). The extensive biochemical literature knowledge embedded in the large pretraining corpora enables language models to grasp biochemical domain-specific concepts. Significant advancements in accuracy and applications have been made for molecule-related tasks, such as biochemical, medical question answering (Taylor et al., 2022; Li et al., 2024; Liu et al., 2023a) and molecule captioning (Liu et al., 2023b; Luo et al., 2024). The field of molecule-text translation plays a crucial role in facilitating efficient molecule characterization and comprehensive understanding for domain experts, particularly admist the rapid expansion of scientific data.

Self-supervised molecular representation learning (MRL) has made significant strides in capturing the properties and functions of small molecules across diverse applications (Guo et al., 2022). This success is built on harnessing various molecular structures, such as 1D SMILES (Simplified Molecular Input Line Entry System) strings (Irwin et al., 2022), 2D graphs (You et al., 2020; Hu et al., 2019; Wang et al., 2022), and 3D conformations (Zhou et al., 2023). Many computational chemistry tasks rely heavily on 2D molecular structures to capture atomic bonding patterns and molecular inter-connectivity (Guo et al., 2022). 2D molecular representation is typically encoded as graph with atoms as nodes and bonds as edges, offering a clear and intuitive depiction of molecular architecture. Nodes are embedded with rich atomic features such as atomic number, formal charge and hybridization state while edges are characterized by bond type, length, and other relevant properties (Duvenaud et al., 2015; Yang et al., 2019). 3D molecular conformers, on the other hand, provide critical

information about the spatial arrangement of atoms. The embedding of atom coordinates directly hint molecular conformation, interactions, and binding affinities in biological systems. Therefore, MRL models have evolved to handle 3D molecular information for downstream tasks that require 3D molecular geometry prediction or generation (e.g. protein-ligand affinity). Nonetheless, each variant of molecular representations contribute uniquely. 1D SMILES provide compact representation of molecular structures, 2D graphs capture the static relationships and connectivity essential for many chemical analyses and 3D structures reflect the dynamic spatial arrangement (Kim et al., 2024; Du et al., 2023).

The success of vision-language modeling methods (Alayrac et al., 2022; Merullo et al., 2022) has accelerated the application of cross-modal alignment in the molecular domain. Studies have adopted contrastive learning (Figure 1A) or the Q-Former (Li et al., 2023) framework (Figure 1B) to align molecular representations with text descriptions (Su et al., 2022; Liu et al., 2023a;b; Li et al., 2024). Q-Former excels in this area due to its effective cross-modal attention and query-based representation. Previous works have aligned only a single view of a molecule within the Q-Former framework (Figure 1B). However, as different dimensions capture distinct molecular characteristics, relying on a single view may be insufficient. For instance, texts describing molecular properties often reference both topology (e.g., ring structures) and spatial conformation (e.g., optimal coordinates). Simultaneous alignment of 2D and 3D views to textual descriptions can resolve ambiguities inherent in a single representation. A simple approach would be to directly align each view to text using two separate alignment modules. However, this leads to several issues. 1) *Separated embedding spaces*. As independent pretrained models or encoders are utilized for 2D and 3D structures, the corresponding embeddings exist in a separate space. Without alignment between the respective multiple views, producing a consistent representation that leverages all information is difficult. 2) *Lack of text consistency*. Cross-modal alignment not only aligns molecular information to text, but also vice versa. Independent utilization of Q-formers lead textual representations to lie in different latent space, which conflicts the purpose of utilization. 3) *High computational cost*. Processing each view independently results in significant computational overhead.

To address these limitations, we propose **Multi-Querying Transformer (MQ-Former)**. MQ-Former approximates the embedding spaces of 2D and 3D structures using a shared self-attention layer and employs a unified text transformer to generate a single, processed text token for each molecule (Figure 1C). Aligning multiple molecular views to the same text provides a more subtle and robust embedding, allowing models to capture both chemical and spatial semantics in a unified representation. In essence, adopting a multi-view approach enables a deeper and more complete molecular understanding. Moreover, by aligning the two views simultaneously, our approach achieves faster training speeds and reduces the training time by more than half compared to handling each view separately.

Our contributions are as follows:

- We incorporate both 2D and 3D molecular structures to guide a more comprehensive understanding of molecules for language models.
- We propose MQ-former, a novel cross-modal projector that can align multiple different views to a unified text embedding space.
- We achieve state-of-the-art performance in molecule-text retrieval and molecule captioning tasks while improving the interpretability of molecular representations. We conduct downstream molecule property question answering and zero-shot molecule editing.

## 2 RELATED WORKS

**Molecule-Text Modeling.** Early approaches utilize 1D SMILES molecular sequences to treat molecules as text sequences by adapting Transformer models (Vaswani, 2017) designed for natural language processing (Irwin et al., 2022; Wang et al., 2019). KV-PLM (Zeng et al., 2022) specifically employs a masked language modeling loss to pretrain on biomedical texts with 1D SMILES representation. MolT5 (Edwards et al., 2022) specializes T5 model (Raffel et al., 2020) and tokenizer for SMILES-to-text and text-to-SMILES translations. Further enhancements represent molecules as 2D graphs. In particular, MoMu (Su et al., 2022) and MoleculeSTM (Liu et al., 2023a) leverage cross-modal contrastive learning to align the molecule graph representation to text. Current approaches to

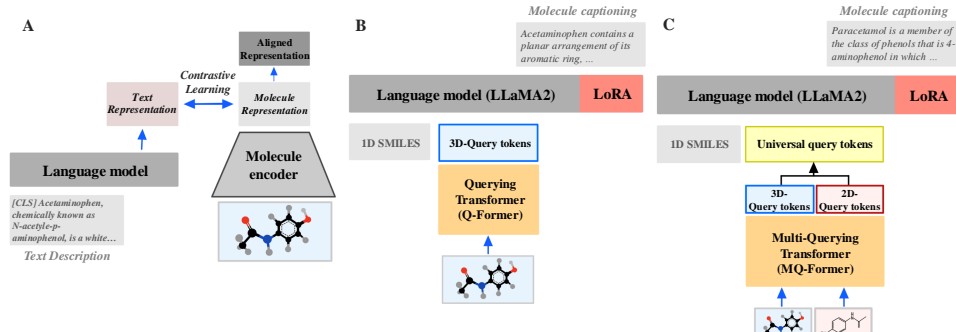

Figure 1: Methods for molecular language modeling

use multi-view representations of molecules primarily rely on contrastive learning, as demonstrated in models like GIT-Mol (Liu et al., 2024) and MolLM (Tang et al., 2024b). Additionally, aided with the development of vision large language models (VLLMs), molecular large language models with multi-modal learning architectures have been developed. Simple projection layers were used in prior works, InstructMol (Cao et al., 2023) and GraphGPT (Tang et al., 2024a), to project molecular graph representations to LLM's input text token space. Recent works have been concentrated on utilizing Q-Former (Li et al., 2023) suggested in vision domain to bridge the gap between molecule and text modality. MolCA (Liu et al., 2023b) and 3D-MoLM (Li et al., 2024) aligns 2D graph and 3D conformer molecular representations to text in purpose to generate effective soft-prompts for large language models. UniMoT (Zhang et al., 2024) employs a vector quantization-driven tokenizer with a Q-Former. Current methods for utilizing multi-view representations of molecules are limited to contrastive learning or usage of specialized tokenizers, failing to achieve simultaneous alignment across all views and text, thereby neglecting the core principle of cross-modal alignment.

## 3 MV-CLAM

MV-CLAM provides molecule captions given multi-view structural information. 2D and 3D molecular structural information is extracted from specialized encoders and processed through MQ-Former's cross-attention layers to update learnable query tokens for each dimension. These query tokens are aligned to textual space via the shared self attention and multi-objective learning, while also considering the alternative view. 2D and 3D queries are combined to create a universal query, which is then passed with the prompt and SMILES strings to the language model for caption generation. The overall framework of MV-CLAM is comprised of three main components: 1) Molecule structural graph encoders for 2D and 3D molecular structures, 2) MQ-Former as a cross-modal projector, and 3) LLaMA2 as the language model. (Figure 2).

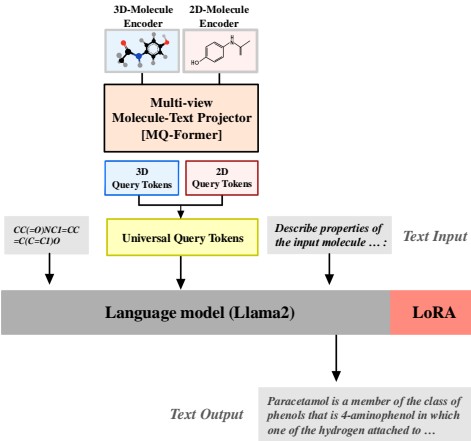

Figure 2: Overall architecture of MV-CLAM. MQ-Former provides universal query which acts as a soft prompt to Llama2, optimized by LoRA

### 3.1 MOLECULAR GRAPH ENCODER

To capture structural information from multiple views, we used molecular embeddings from both 3D and 2D structural encoders. For the 3D encoder $f_{3d}$, we deployed **Uni-Mol** (Zhou et al., 2023), a SE(3)-transformer based model pretrained on 209 million 3D molecular conformations using two tasks: 3D position recovery and masked atom prediction. Input 3D molecule for Uni-Mol is denoted as $m_{3d} = (\mathcal{V}, \mathbf{f}, \mathbf{P})$, where $\mathcal{V}$ and $\mathbf{f}$ each represents atomic nodes and their features, and $\mathbf{P} \in \mathbb{R}^{|\mathcal{V}| \times 3}$ represents 3D coordinates of atoms. Pair representations are initialized by invariant spatial positional encoding from atom coordinates and interact

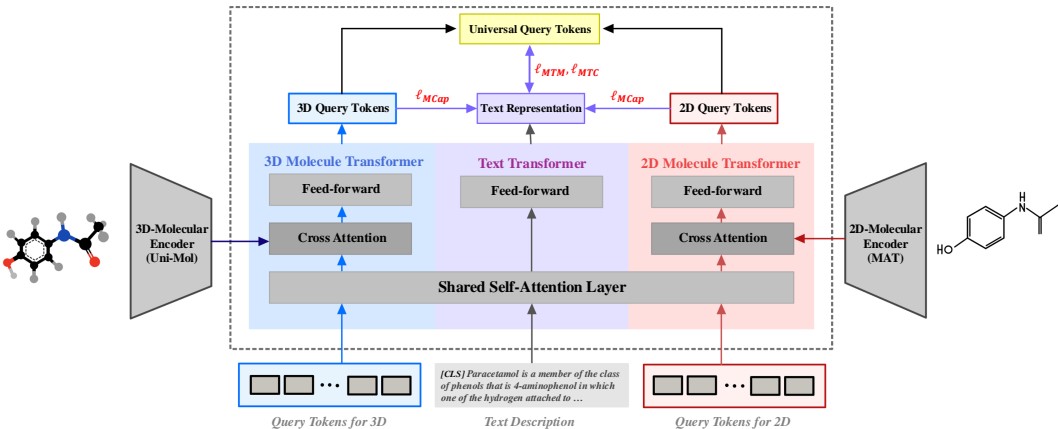

Figure 3: Training scheme of MQ-Former

with atom representations. The output atomic representation $H_{3d} \in \mathbb{R}^{|\mathcal{V}| \times d_{3d}}$, where $h_i$ corresponds to the $i$-th atom and $d_{3d}$ denotes hidden dimension size of $H_{3d}$, updates learnable 3D query tokens through the cross-attention layers in MQ-Former's 3D molecular transformer block.

$$H_{3d} = [h_1, h_2, ..., h_{|\mathcal{V}|}] = f_{3d}(m_{3d}) \tag{1}$$

For the 2D molecular encoder $f_{2d}$, we adopted **Molecule Attention Transformer (MAT)** (Maziarka et al., 2020), pretrained on two million molecule samples from ZINC15 dataset (Maziarka et al., 2020). Given 2D molecule $m_{2d} = (\mathcal{V}, \mathbf{f}, \mathbf{A})$ where $A$ represents edges within the molecule as adjacency matrix, MAT generates atomic representations $H_{2d} \in \mathbb{R}^{|\mathcal{V}| \times d_{2d}}$ using a specialized molecule-specific attention mechanism that considers edges, atomic distances and atomic features. The atomic representations interact with the learnable 2D query tokens via cross-attention layers in 2D molecular transformer block.

$$H_{2d} = [h_1, h_2, ..., h_{|\mathcal{V}|}] = f_{2d}(m_{2d}) \tag{2}$$

### 3.2 MQ-Former: Multi-Querying Transformer

Previous studies applying Q-Former to the molecular domain projects single-dimensional structural embeddings into the textual space (Li et al., 2024; Zhang et al., 2024). These models consist of a single molecule transformer and a text transformer. However, this approach is inherently limited in its capacity to handle more than two modalities. MQ-Former addresses the limitation by introducing a novel architecture capable of aligning multiple modalities to the text space (Figure 3). Our approach combines structural representations of two dimensions, but the architecture can be extended using multiple molecule transformers and a single text transformer. Each molecule transformer, based on the BERT architecture with additional cross-attention layer, processes $K$ learnable query tokens specific to their respective views. Following previous studies (Li et al., 2024; Liu et al., 2023b), we adopt the SciBERT (Beltagy et al., 2019) architecture for the text transformer and initialize all blocks with SciBERT's pretrained weights. Hence, textual descriptions $S$ of length $L$ are tokenized with SciBERT's tokenizer $f_{sci}$ to $X_{text}$ before being processed through MQ-Former's text transformer. The cross-attention mechanism extracts relevant information from embeddings into the query tokens, and shared self-attention layers enable information exchange across text and multi-view data.

Figure 3 illustrates MQ-Former generating a universal query tokens for a molecule given two different views. Two molecule transformer modules each updates distinct $K$ query tokens $Q_{2d} \in \mathbb{R}^{K \times 768}$ and $Q_{3d} \in \mathbb{R}^{K \times 768}$, which are randomly initialized. The learned query tokens, $\hat{Q}_{2d}$ and $\hat{Q}_{3d}$ of same size, are updated representations of these initial tokens, refined through the alignment of multiple molecule views and textual descriptions $X_{text} \in \mathbb{R}^{L \times 768}$. Updated query tokens are concatenated to create a single universal query $\hat{Q} \in \mathbb{R}^{2K \times 768}$, containing complementary structural information aligned to textual space. The resulting universal query tokens are then used as inputs for the

language model, along with 1D SMILES string and task prompt as depicted in Figure 2.

$$\hat{Q} = f_{\text{concat}}(\hat{Q}_{2d}, \hat{Q}_{3d}) = f_{\text{MQformer}}(H_{2d}, H_{3d}, X_{\text{text}}, Q_{2d}, Q_{3d}) \tag{3}$$

## 3.3 LLAMA2 & LORA

The pretraining corpus of LLaMA2 (Touvron et al., 2023) includes a vast amount of biomedical literature and thereby exerts powerful text generation capability with internal chemistry knowledge. This allows LLaMA2 to effectively interpret 1D molecular sequences and address tasks related to molecular comprehension. The language model adopts a causal mask to generate textual responses, where the prediction of each token depends on the preceding tokens. For the final prediction, each token is mapped to the most probable word in vocabulary using a softmax function after a linear layer. Despite its inherent capabilities, the language model necessitates fine-tuning to effectively address the universal queries posed by MQ-Former, particularly due to the modifications in the tokenizer resulting from changes in module processing of textual descriptions. To facilitate efficient fine-tuning, we implemented low-rank adaptation (LoRA, Hu et al. (2021)).

## 4 TRAINING MV-CLAM

The training of MV-CLAM consists of two stages. 1) Guiding MQ-Former to align both multi-view molecular representations to textual space, and 2) Refining query tokens as soft prompts to be effectively utilized by LLaMA2. Molecular encoders are frozen during the entire pipeline.

### 4.1 STAGE 1: TRAINING MQ-FORMER

Two sets of $K$ learnable query tokens are updated by each molecule transformer block in Stage 1. Molecule transformer blocks hold self-attention, cross-attention and feed-forward layers. Specifically, the self attention layers in all blocks of MQ-Former are shared to exchange information between modalities and view. The objective is to train MQ-Former to better align molecular representations given by cross-attention to textual space. The training employs a multi-objective training loss constituted of molecule-text contrasting $\ell_{MTC}$, molecule-text matching $\ell_{MTM}$ and molecule captioning $\ell_{MCap}$ inspired by the BLIP-2 framework (Li et al., 2023; 2024).

**Molecule-text Contrasting**. During $\ell_{MTC}$ computation, uni-modal self-attention mask ensure each transformer processes query tokens independently, preventing information exchange and promoting distinct representations for matching and non-matching molecule-text pairs. The 2D and 3D query tokens $Q_{2d}(i)$, $Q_{3d}(i)$ for $i$-th molecule are processed through their respective molecule transformers. Our $2K$ universal query token $\hat{Q}(i)$ is formed by concatenating the learned query sets.

$\ell_{MTC}$ is measured as cosine similarity between the universal query $\hat{Q}(i)$ and text representation $X_{\text{text}}(i)$ with temperature scaling for precision. $\ell_{MTC}$ is computed as the batch mean of the sum of the molecule-to-text loss $\ell_{g2t}$ and text-to-molecule loss $\ell_{t2g}$. $\ell_{g2t}$ encourages the universal query representation which encodes both 2D and 3D molecular structures, to match its corresponding text representation while contrasting it against all other text representations within the batch. Similarly, $\ell_{t2g}$ aligns the text representation with its matching molecular query. Together $\ell_{MTC}$ form a bidirectional alignment between molecular features and textual descriptions, enhancing the ability of MQ-Former to jointly represent and contrast molecules and their associated textual descriptions. $\ell_{g2t}$ and $\ell_{t2g}$ is as written below, where $M$ is the size of the batch and $\tau$ is the temperature parameter.

$$\ell_{g2t} = \sum_{i=1}^{M} \log \frac{\exp(\max_k \cos(\hat{Q}(i), X_{\text{text}}(i))/\tau)}{\sum_{j=1}^{M} \exp(\max_k \cos(\hat{Q}(i), X_{\text{text}}(j))/\tau)}$$

$$\ell_{t2g} = \sum_{i=1}^{M} \log \frac{\exp(\max_k \cos(X_{\text{text}}(i), \hat{Q}(i))/\tau)}{\sum_{j=1}^{M} \exp(\max_k \cos(X_{\text{text}}(i), \hat{Q}(j))/\tau)}$$

$$\tag{4}$$

**Molecule-text Matching**. $\ell_{MTM}$ is for a binary classification task to predict matching molecule-text pairs. Bi-directional self-attention masks lead all text and molecular embeddings from different dimensions to share their information, guiding MQ-Former to capture fine-grained similarities between the domains. Universal query tokens are obtained then processed through a linear classifier

after mean pooling. Let $\rho(\hat{Q}(i), X_{\text{text}}(i))$ denote the predicted probability that universal query $\hat{Q}(i)$ matches its corresponding text description $X_{\text{text}}(i)$. $\ell_{MTM}$ is calculated as follows:

$$\ell_{MTM} = \frac{1}{M} \sum_{i=1}^{M} \Big( -\log \rho(\hat{Q}(i), X_{\text{text}}(i)) + \log \rho(\hat{Q}(i), X_{\text{text}}(j)) + \log \rho(\hat{Q}(r), X_{\text{text}}(i)) \Big) \quad (5)$$

where $X_{\text{text}}(j)$, $\hat{Q}(r)$ are randomly selected negative samples from the batch. Overall, $\ell_{MTM}$ aids MQ-Former to maximize the likelihood of matched pairs and minimize mismatches, enhancing its ability to differentiate between true and false pairs.

**Molecule Captioning**. $\ell_{MCap}$ is designed to generate accurate text descriptions based on multi-view query tokens. A multi-modal causal self-attention masking strategy ensures that molecule query tokens rely on cross-attention with molecular embeddings for text generation, preventing direct access to text tokens. Text is generated auto-regressively, where each token is predicted sequentially based on the corresponding molecular queries. Instead of harnessing universal queries, $\ell_{MCap}$ sums up separate losses for 2D and 3D query tokens, ensuring that each query token retains its unique dimensional information while improving the captioning ability. The $\ell_{MCap}$ is defined as follows:

$$\ell_{MCap} = -\frac{1}{M} \sum_{i=1}^{M} \log p(X_{\text{text}}(i)|\hat{Q}_{2d}(i)) - \frac{1}{M} \sum_{i=1}^{M} \log p(X_{\text{text}}(i)|\hat{Q}_{3d}(i)) \quad (6)$$

where $p(X_{\text{text}}|\hat{Q}_{2d})$ and $p(X_{\text{text}}|\hat{Q}_{3d})$ represents the probability of generating the text description based independently on 2D or 3D molecular queries, respectively. While the other two losses focus on aligning or matching molecule-text pairs, the $\ell_{MCap}$ directly impacts the ability to generate new text based on molecular representations. Given its critical role, we assigned a greater weight $\alpha$ during multi-objective training, guiding MQ-Former to generate quality query tokens for text-generation tasks. Overall, the total loss for training MQ-Former $\ell_{MQ}$ in Stage 1 is as follows:

$$\ell_{MQ} = \ell_{MTC} + \ell_{MTM} + \alpha * \ell_{MCap} \quad (7)$$

### 4.2 STAGE 2: SPECIALIZING LLaMA2 FOR MOLECULE CAPTIONING

In Stage 2, MQ-Former is further trained alongside LLaMA2 to generate molecular descriptions. The goal is to enhance MQ-Former's ability to produce universal queries that are not only aligned with the textual space but better interpretable by LLaMA2. In this stage, textual descriptions are tokenized and decoded using LLaMA tokenizer. MQ-Former is fine-tuned using $\ell_{MTC}$ and $\ell_{MTM}$ and the captioning loss is derived from output captions of LLaMA2. Universal query tokens, 1D SMILES are given as input with prompt. LoRA (Hu et al., 2021) is employed for efficient finetuning, focusing on a subset of parameters. Detailed LoRA setting are in Appendix A3.

## 5 EXPERIMENTS

### 5.1 DATASETS

**PubChem324K**. For molecule-text alignment and molecule captioning, we collected 324k molecular SMILES-text pairs from PubChem (Kim et al., 2021). 2D graph features were constructed using Maziarka et al. (2020), and 3D conformers were generated with ETKDG and optimized using the MMFF algorithm in RDKit (Landrum et al., 2013). We follow dataset construction as provided in 3D-MoLM (Li et al., 2024) which also requires 3D molecular conformations. High-quality subset of 15k pairs with text longer than 19 words are sampled for train, valid, test datasets. Shorter pairs are used for pretraining. The statistics for the final PubChem324k dataset used in this study are presented in Appendix Table 6.

### 5.2 BENCHMARK MODELS

Baseline models include 1) pretrained language models for science: Sci-BERT (Beltagy et al., 2019), 2) models with molecule-text contrastive learning: KV-PLM (Zeng et al., 2022), MoMu (Su et al., 2022), MoleculeSTM (Liu et al., 2023a) and 3) models with Q-Former modules: MolCA (Liu et al.,

Table 1: Molecule-Text retrieval performance in batch and test set for different models. The highest value in each category is indicated in bold, and the second highest value is underlined. For MoleculeSTM* and MolCA*, we report results from UniMoT (Zhang et al., 2024).

| Model | Retrieval in batch | | | | Retrieval in test set | | | |
| | M2T | | T2M | | M2T | | T2M | |
| | ACC | R@20 | ACC | R@20 | ACC | R@20 | ACC | R@20 |
|---|---|---|---|---|---|---|---|---|
| **1D SMILES** | | | | | | | | |
| Sci-BERT(Beltagy et al., 2019) | 85.32 | 98.74 | 84.20 | 98.43 | 41.67 | 87.31 | 40.18 | 86.77 |
| KV-PLM(Zeng et al., 2022) | 86.05 | 98.63 | 85.21 | 98.47 | 42.80 | 88.46 | 41.67 | 87.80 |
| **2D Graph** | | | | | | | | |
| MoMu-S(Su et al., 2022) | 87.58 | 99.24 | 86.44 | 99.38 | 47.29 | 90.77 | 48.13 | 89.92 |
| MoMu-K(Su et al., 2022) | 88.23 | 99.41 | 87.29 | 99.42 | 48.47 | 91.64 | 49.46 | 90.73 |
| MoleculeSTM* (Liu et al., 2023a) | 90.50 | 99.60 | 88.60 | 99.50 | 52.70 | 92.90 | 53.20 | 92.50 |
| MolCA* (Liu et al., 2023b) | 92.60 | 99.80 | 91.30 | 99.50 | 67.90 | 94.40 | 68.60 | 93.30 |
| **2D Graph + Tokenizer** | | | | | | | | |
| UniMoT(Zhang et al., 2024) | 93.60 | **100.0** | 92.70 | 99.40 | 69.50 | **96.30** | 69.80 | 94.40 |
| **3D Conformer** | | | | | | | | |
| 3D-MoLM(Li et al., 2024) | 93.50 | **100.0** | 92.89 | 99.59 | 69.05 | 95.91 | 70.13 | 94.88 |
| **2D Graph + 3D Conformer** | | | | | | | | |
| MV-CLAM | **96.57** | 99.95 | **97.03** | **99.95** | **76.32** | 96.57 | **77.03** | **96.42** |

2023b), 3D-MoLM (Li et al., 2024), UniMoT (Zhang et al., 2024). For molecule captioning, we also benchmark Llama2-7B and 2D-MoLM, each as a variant of 3D-MoLM using 1D and 2D information along with MolT5 (Edwards et al., 2022) and InstructMol (Cao et al., 2023).

# 6 RESULTS

## 6.1 MOLECULE-TEXT RETRIEVAL

We evaluate MV-CLAM for molecule-text retrieval on the PubChem324k dataset. After pretraining for 35 epochs, the model is fine-tuned on the training subset with longer captions for 10 epochs. We perform two rounds of evaluation on molecule-to-text and text-to-molecule retrieval tasks, using Accuracy and Recall@20 metrics: within batch size of 64 and is across the entire test set. We report baseline performances as written in literature (Li et al., 2024; Zhang et al., 2024).

As shown in Table 1, MV-CLAM outperforms baseline approaches that represent molecules as 1D SMILES strings, 2D graphs, or 3D conformers. Additionally, results are achieved within a total of 45 epochs, comparative to 3D-MoLM that trains for 60 epochs. We attribute our superior performance to 1) our usage of unified query that aligns both 2D and 3D information to text and 2) modification on the Q-Former's multi-objective loss to amplify molecule captioning loss. As a result, the text transformer is better equipped to decode molecule descriptions under 2D and 3D conditions, benefiting from the enriched molecular information. While good retrieval performance is often indicative of strong cross-modal understanding that benefit captioning tasks as demonstrated in previous studies (Li et al., 2024; 2023), the relationship is not absolute. Hence we proceed to evaluate the performance of molecule captioning.

## 6.2 MOLECULE CAPTIONING

Following previous studies(Li et al., 2024), we use BLEU, ROUGE, METEOR metrics to evaluate molecule captioning on the PubChem324k dataset. As outlined in Section 4.2, we apply LoRA to fine-tune LLaMA2 for the molecular domain, training 10 epochs on the pretraining subset and an additional 10 epochs on the training subset. Table 2 shows MV-CLAM consistently outperforms all baselines. Given that the PubChem324k dataset include molecular nomenclature, our model excels not only in generating appropriate captions based on molecular structure including information on clinical usage and chemical properties but also in accurately predicting molecular names. Appendix Table 8 highlights the model's ability to correctly identify International Union of Pure and Applied Chemistry (IUPAC) nomenclature and generic drug names. These two types of nomenclature differ significantly in terms of language model processing. IUPAC names follow systematic chemical

Table 2: Molecule captioning performance across models. The highest value in each category is bolded, and the second highest is underlined. Models marked with †were pretrained on larger datasets, as noted in their original papers. Results for InstructMol and MolCA are from UniMoT (Zhang et al., 2024), with MolCA evaluated in two variations using OPT-125M (small) and OPT-1.3B (large) as language models.

| | BLEU-2 | BLEU-4 | ROUGE-1 | ROUGE-2 | ROUGE-L | METEOR |
|---|---|---|---|---|---|---|
| **1D SMILES** | | | | | | |
| MolT5-Small(Edwards et al., 2022) | 22.53 | 15.23 | 30.44 | 13.45 | 20.30 | 23.98 |
| MolT5-Base(Edwards et al., 2022) | 24.51 | 16.61 | 32.19 | 14.04 | 21.35 | 26.10 |
| MolT5-Large(Edwards et al., 2022) | 25.87 | 17.28 | 34.07 | 16.42 | 23.41 | 28.04 |
| Llama2-7B†(Li et al., 2024) | 27.01 | 20.94 | 35.76 | 20.68 | 28.88 | 32.11 |
| **2D Graph** | | | | | | |
| MoMu-Small(Su et al., 2022) | 22.86 | 16.01 | 30.98 | 13.65 | 20.75 | 24.35 |
| MoMu-Base(Su et al., 2022) | 24.74 | 16.77 | 32.45 | 14.62 | 22.09 | 27.16 |
| MoMu-Large(Su et al., 2022) | 26.34 | 18.01 | 34.75 | 16.86 | 24.76 | 28.73 |
| 2D-MoLM†(Li et al., 2024) | 27.15 | 21.19 | 36.02 | 20.76 | 29.12 | 32.28 |
| InstructMol*(Cao et al., 2023) | 18.90 | 11.70 | 27.30 | 11.80 | 17.80 | 21.30 |
| MolCA-Small*(Liu et al., 2023b) | 25.90 | 17.50 | 34.40 | 16.60 | 23.90 | 28.50 |
| MolCA-Large*(Liu et al., 2023b) | 28.60 | 21.30 | 36.20 | 21.40 | 29.70 | 32.60 |
| **2D Graph + Tokenizer** | | | | | | |
| UniMoT(Zhang et al., 2024) | 31.30 | 23.80 | 37.50 | 23.70 | 33.60 | 34.80 |
| **3D Conformer** | | | | | | |
| 3D-MoLM(Li et al., 2024) | 30.32 | 22.52 | 36.84 | 22.32 | 31.23 | 33.06 |
| **2D Graph + 3D Conformer** | | | | | | |
| MV-CLAM | **31.75** | **24.48** | **40.43** | **25.72** | **33.79** | **36.54** |

rules, making them complex and highly structured, while generic drug names are more standardized and commonly used in clinical contexts. Despite these differences, MV-CLAM successfully identifies both types of names, showcasing its ability to handle a range of linguistic and chemical complexities. Moreover, MV-CLAM demonstrates its capacity to generate literature-matching captions absent in ground truth, as seen in the case of *Rifapentine* in Appendix Table 8, highlighting the ability to produce highly informed and contextually relevant outputs.

### 6.3 EFFECTIVENESS OF MQ-FORMER

In this section, we substantiate the effectiveness of incorporating multi-view chemical information within the MQ-Former architecture. We conduct both quantitative and qualitative analysis to compare our superiority to the usage of single-view molecule representation with Q-Former: 2D-QFormer and 3D-QFormer. Molecular encoders are identically set for the ablation studies.

As a quantitative analysis, we show that the combination of both modalities leads to a notable synergistic effect, improving the model's overall performance (Table 3). By combining the two perspectives, the model gains a richer understanding of molecular properties which in turn improves accuracy and expressiveness of molecule captioning. The alignment of both modalities ensures that critical information is utilized, leading to more robust and detailed predictions, supporting the hypothesis that well-orchestrated multi-modal fusion can surpass the limitations of single-modal approaches in capturing complex molecular characteristics. Additionally, we conducted an ablation experiment utilizing multi-view molecular embeddings within a single Q-Former module described in Section A.4.4, which further highlights the benefits of MQ-Former.

We exemplify two case studies to interpret how each transformer module and modality focus on distinct aspects of the molecule and its corresponding text. These qualitative studies provide insight into the alignment process by analyzing how different views contribute to the comprehensive understanding of molecular structures and their textual descriptions.

Table 3: Molecule Captioning Ablation Study

| | BLEU-2 | BLEU-4 | ROUGE-1 | ROUGE-2 | ROUGE-L | METEOR |
|---|---|---|---|---|---|---|
| 2D-Qformer | 29.72 | 22.26 | 38.22 | 23.45 | 31.61 | 34.22 |
| 3D-Qformer | 29.45 | 22.03 | 37.86 | 23.11 | 31.83 | 33.79 |
| Ours | **31.75** | **24.48** | **40.43** | **25.72** | **33.79** | **36.54** |

**Case Study 1: Visualizing Attention Maps for 2D and 3D Query Tokens.** Embedding grounded on different latent spaces and dimensions differently align molecular information to text. Visualization of the distinct alignment is performed by extracting and comparing the attention maps of the shared self-attention layers when processing 2D and 3D query tokens respectively with text tokens.

In the first example, only 2D queries assign exceptionally high attention weights to the word '*water*' (Appendix Figure 5). The discrepancy between two attention maps implies that 2D query tokens efficiently focus on chemical and material properties that may be neglected in 3D settings. In contrast, for the sentences containing of structural equation information, 3D attention map shows strong attention to positions inherent in molecular formula (Appendix Figure 6). Significant attention is assigned on the number '*3*' in 3D attention map, less pronounced in the 2D attention map. This suggests that the 3D query tokens, informed by 3D spatial coordinates, are more attuned to the structural aspects of the molecule. In summary, 2D and 3D query tokens each focus on different aspects within the same sentence, complementing each other to prevent critical information from being missed and thereby leading to more informative and accurate molecule descriptions.

**Case Study 2: Comparing molecule captions with 2D-Qformer and 3D-Qformer.** We illustrates the difference in captioning results between the uni-modal Q-Former ablation models and ours demonstrating the effects of utilizing multi-view molecular understanding in text generation (Appendix Figure 8). The 2D and 3D uni-modal ablations struggle to fully capture complex and large structures like '*(R)-3-hydroxytriacontanoyl-CoA*'. The ablation models fail to retain sufficient structural information required to differentiate long carbon chains with their functional groups. However, our model captures not only carboxylic acid but also phosphonate groups, which are often considered bioisosteric replacements for sulfonate acids in medicinal chemistry due to their structural similarity (Macchiarulo & Pellicciari, 2007). In comparison, the ablation models only managed to capture one of these groups, indicating that multi-view approach enables the generation of accurate nomenclature and richer descriptive information.

## 6.4 MOLECULAR QUESTION-ANSWERING

For the molecular question-answering task, we utilized the 3D-MoIT (Li et al., 2024) dataset, which includes question-prompt and text-answer pairs derived from the same PubChem data we used in prior. Dataset statistics are in Appendix Table 7 The dataset consists of three distinct subsets: 1) Question-answering about non-3D properties, 2) Question-answering about 3D properties, and 3) Descriptive molecular properties. To fine-tune MV-CLAM for this task, we initialized the model using Stage 2 (molecule captioning) checkpoints and further trained it on the 3D-MoIT dataset. For computed property prediction, we evaluated performance using mean absolute error (MAE). For descriptive property prediction, we measured BLEU, ROUGE, and METEOR scores.

For baselines, we reproduced results for 3D-MoLM and 2D-MoLM (with MAT (Maziarka et al., 2020) graph encoder). These baselines represent single-modal alignment using Q-Former, and provides a fair point of comparison to demonstrate the efficacy of our multi-view cross-modal alignment. Tables 4 and 5 show that MV-CLAM consistently outperformed the single-modal models.

Table 4: Comparison of Descriptive Property Generation Performance

| Model | BLEU-2 | BLEU-4 | ROUGE-1 | ROUGE-2 | ROUGE-L | METEOR |
|---|---|---|---|---|---|---|
| 2D-MoLM | 31.24 | 25.13 | 39.30 | 25.16 | 34.11 | 49.88 |
| 3D-MoLM | 29.22 | 22.82 | 37.38 | 22.54 | 31.47 | 27.29 |
| Ours | **31.70** | **25.60** | **39.61** | **25.46** | **34.51** | **50.61** |

Table 5: Comparison of Q&A performance on 3D and non-3D properties

| Model | Molecular Weight | LogP | Complexity | Topological Polar Surface Area | HOMO | LUMO | HOMO-LUMO | SCF Energy |
|---|---|---|---|---|---|---|---|---|
| 2D-MoLM | 47.51 (0.98) | 0.89 (0.99) | 110.78 (0.99) | 16.65 (0.99) | 0.78 (0.99) | 0.47 (0.99) | 0.39 (0.90) | 0.98 (1.00) |
| 3D-MoLM | 42.76 (0.96) | 1.25 (0.96) | 105.03 (0.96) | 20.97 (0.92) | 0.42 (0.99) | 0.44 (0.98) | 1.26 (0.99) | 1.22 (0.98) |
| Ours | **21.35 (0.92)** | **0.69 (0.94)** | **55.14 (0.91)** | **9.65 (0.91)** | **0.35 (0.98)** | **0.42 (0.93)** | **0.35 (0.99)** | **0.32 (0.99)** |

## 6.5 ZERO-SHOT MOLECULE EDITING

Unlike conventional natural languages, SMILES encode molecular topology and properties demanding a specialized understanding of its notation system. Thereby, previous efforts in text-based de-

novo molecule generation with large language models typically involves training or developing tokenizers that account for the unique grammar of SMILES (Edwards et al., 2022). In contrast, our approach is the first to attempt generating SMILES directly using the raw LLaMA tokenizer. By fine-tuning MV-CLAM, we enabled the model to output SMILES strings without additional tokenizer training. Initialized with the Stage 2 checkpoint, the model was trained to generate target SMILES sequences based on the universal molecular queries produced by MQ-Former. Following this training, we conducted zero-shot molecule editing, utilizing the model's pre-existing multi-view molecular understanding from prior stages. We evaluate the edited results by computing desired chemical properties using RDKit (Landrum et al., 2013).

In this section we show successful case studies of the language model generating valid SMILES strings with adequate property modifications. Compared to previous works which mostly generate mere modifications of a single functional group, MV-CLAM generates diversified chemical structure modifications that may not be immediately obvious. This ability to generate more complex modifications is particularly advantageous for domain experts, as simple functional group changes are typically easy to perform manually. We attribute this diversity to the model's robust understanding of molecules within the textual space. The alignment between molecules and text is achieved by focusing on distinct substructures and molecular properties through the multi-view approach. Additional examples and more details in the training procedure can be found in Appendix A.5.

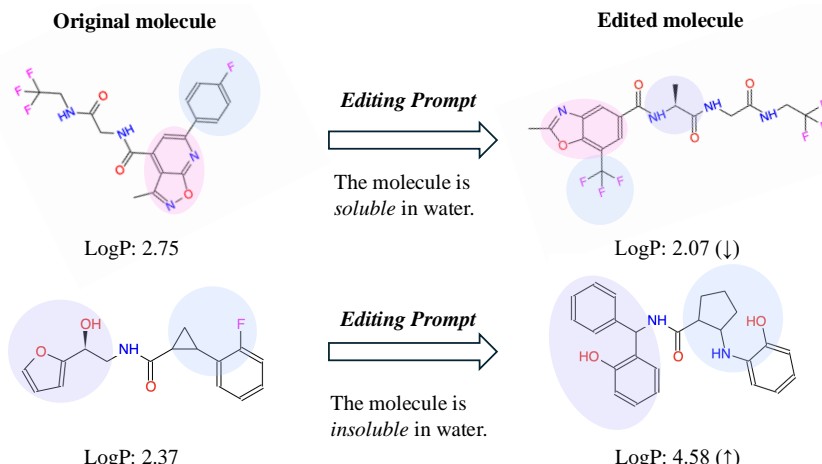

Figure 4: Zero-shot editing with chemical properties

## 7 CONCLUSION

In this paper, we introduce MV-CLAM equipped with MQ-Former, a novel cross-modal projector. The essence of cross-modal projection lies in aligning the enriched molecular representation spaces with the text space of language models. Our architecture successfully retains complementary information from multiple dimension into a single universal token easily interpreted by large language models for molecule description tasks. Extensive experiments demonstrate that MV-CLAM has successfully fine-tunes large language models for molecule understanding, including molecule-text retrieval and molecule captioning tasks, with potential for broader applications.

For future work, we aim to extend this framework to incorporate additional molecular representations, including 1D chemical structures, proteomics, and multiomics data. By aligning more views within MV-CLAM's architecture, we anticipate improved navigation of the drug space and a deeper understanding of molecular interactions across biological contexts. Additionally, curating larger molecule-text datasets is expected to enhance the model's performance and its ability to generalize to subtle molecular variations.

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

# A APPENDIX

## A.1 RELATED WORKS

**Molecular representation learning.** Recent research in representation learning for molecules has seen significant advancements, particularly in leveraging large-scale unlabeled molecular data. SMILES-BERT (Wang et al., 2019), MolBERT (Li & Jiang, 2021) adapts the BERT architecture on SMILES string for molecular property prediction tasks. To better focus on structural information of molecules, various graph-based representation learning models were presented. MolCLR (Wang et al., 2022) specifically tailored contrastive learning for molecular graphs using data augmentation while MAT (Maziarka et al., 2020) reinterpreted the attention mechanism of transformers to consider distance and edges. More recent works concentrate on employing 3D geometry, mostly to exploit 3D spatial coordinates. GraphMVP (Liu et al., 2021) proposed a contrastive learning framework that bridges 2D topological and 3D geometric views of molecules. GEM (Fang et al., 2022) incorporated 3D geometric information by using bond angles and lengths as additional edge attributes in molecular graphs. Uni-Mol is a SE(3)-transformer based model pretrained via 3D position recovery and masked atom prediction. Additionally, MolFormer (Wu et al., 2023) integrates SMILES, graph, and 3D conformer information in a unified transformer architecture for molecular property prediction. These recent advancements demonstrate a trend towards incorporating more diverse and rich molecular information to improve the quality and applicability of learned representations, validating the approach of our research.

## A.2 DATASETS STATISTICS

**PubChem.** We gathered 324k SMILES-text pairs from PubChem, generating 2D graphs and 3D conformations using existing methods (Maziarka et al., 2020; Landrum et al., 2013). Molecules with valid structures were used, with 15k longer-text pairs for training, and shorter ones for pretraining.

Table 6: PubChem324k dataset statistics

| Subset | #Molecule-Text Pairs | #Min Words | #Avg Words |
|---|---|---|---|
| Pretrain | 290,507 | 1 | 17.84 |
| Train | 11,753 | 20 | 57.24 |
| Valid | 977 | 20 | 58.31 |
| Test | 1,955 | 20 | 55.21 |

For the molecule captioning task, we chose not to use ChEBI-20 dataset (Degtyarenko et al., 2007) due to two main considerations (Li et al., 2024). First, ChEBI-20 is a curated subset of PubChem, which introduces potential issues of data redundancy and leakage given the overlap between the two datasets. Second, ChEBI-20 replaces molecular names with generic terms like 'the molecule', limiting the evaluation of the model's ability to associate structural features with accurate molecular names. Therefore, we utilized the PubChem dataset, which retains molecular names and offers a broader variety of structures, ensuring a more comprehensive evaluation of our framework in molecule captioning task.

**ZINC20.** Following the experiment settings of Liu et al. (2023a), 200 molecules randomly selected from the ZINC20 dataset are given 6 single-objective molecule editing instructions. The 200 molecules follow the property distribution of the entire dataset, and do not overlap with the PubChem324k training dataset in previous stages. The six instructions are the following. 1) The molecule is soluble in water. 2) The molecule is insoluble in water. 3) The molecule has high permeability. 4) The molecule has low permeability. 5) The molecule is like a drug. 6) The molecule is not like a drug. 7) The molecule has more hydrogen bond donors. 8) The molecule has more hydrogen bond acceptors.

**3D-MoIT.** A total of 18439K molecule-instruction text pairs are employed using the dataset split as given in the original paper (Li et al., 2024). The dataset consists of two types of molecular property prediction tasks: (1) Computed property prediction including 3D-dependent properties (e.g. HOMO) and (2) descriptive property prediction.

Table 7: Statistics of the PubChemQC and PubChem datasets across different subsets.

| Subset | PubChemQC | | PubChem | | |
|---|---|---|---|---|---|
| | #Mol | #Comp. QA | #Mol | #Comp. QA | #Desc. QA |
| Pretrain | 3,119,717 | 12,478,868 | 301,658 | 1,199,066 | 1,508,290 |
| Train | 623,944 | 2,495,776 | 12,000 | 46,680 | 60,000 |
| Valid | 77,993 | 311,972 | 1,000 | 3,898 | 5,000 |
| Test | 77,993 | 311,972 | 2,000 | 7,785 | 10,000 |

## A.3 EXPERIMENTAL SETTINGS

**Stage 1 Molecule-Text Retrieval Pretraining**. Stage 1 serves to effectively transform molecular representations into query tokens interpretable in textual space. Using the PubChem324k pretraining subset with shorter textual descriptions, that is less informative but easier to align, MQ-former is trained for 35 epochs. A total of 301,658 molecules generated valid 2D graphs and 3D conformers, and thereby was used for pretraining. The goal of this stage was to optimize MQ-Former's universal query generation by multi-objective training (molecule-text contrasting, molecule-text contrasting, and molecule captioning). Pretraining was conducted for 35 epochs using 3 NVIDIA A6000 GPUs with a batch size of 99. Learnable query tokens of each view was set to 12 tokens and were randomly initialized. Both the Uni-Mol and MAT graph encoders were frozen throughout the pipeline to prevent the model from focusing too much on modifying the graph encoders, ensuring the training prioritized aligning representations with the textual space. To put emphasis on the decoding ability given the molecule tokens, we assigned a weight of 2 to the captioning loss. Maximum text length was configured to 256. We used an optimizer with a warmup step of 200 and a learning rate scheduler with a decay rate of 0.9. Gradient accumulation was set to 1 batch per step.

**Stage 1 Molecule-Text Retrieval Finetuning**. After 35 epochs of pretraining, we loaded the checkpoint and fine-tuned MQ-Former for an additional 10 eopchs on PubChem's train, validation and test datasets, consisting of 12,000, 1,000, and 2,000 molecules respectively. This serves to raise alignment capability given longer and more complex textual descriptions. The optimizer, learning rate scheduler, batch size and text length settings are identical to the previous phase.

**Stage 2 Molecule Captioning Pretraining**. Stage 2 serves to further refine the universal tokens in a manner suited to a specific language model, LLaMA2 (Touvron et al., 2023) available at `https://huggingface.co/baffo32/decapoda-research-llama-7B-hf`. Using the trained model checkpoint from Stage 1 training stage, we conducted 10 epochs of pretraining on the PubChem dataset. During the phase, we optimized two tasks: molecule-text contrasting and molecule-text matching for MQ-Former, while using LLaMA2 for the molecule captioning task. The universal query generated by MQ-Former, along with the 1D SMILES string and an instruction prompt were given as input to the language model to generate textual descriptions for the molecules.

To fine-tune LLaMA2 efficiently, we employed LoRA (Hu et al., 2021) with a configuration of $r=8$, $\alpha=32$, and a 0.1 dropout rate. These settings were applied to the $[k_{proj}, v_{proj}, q_{proj}, o_{proj}, gate_{proj}, up_{proj}, down_{proj}]$ modules, adding 19 million trainable parameters, which constituted 0.29% of the total parameters in the LLaMA2-7B model. Unlike Stage 1, we used batch size of 30 with a maximum text length of 320 considering the prompt size. Token length for generation was set to range between 128 and 320. Gradient accumulation was set to 2. The training was carried out using 3 NVIDIA A6000 GPUs.

**Stage 2 Molecule Captioning Fine-tuning**. Stage 2 pretraining checkpoint was further finetuned on the train-validation-test datasets. Experimental settings are identical with stage 2 pretraining phase, excluding batch size which was reduced to 18.

**Downstream Tasks: Question Answering**. For robust guidance into instruction tuning, the three sub-datasets of 3D-MolT Li et al. (2024) were used in combination for training a single epoch. The pretrained MV-CLAM checkpoints from the molecule captioning stage were used for initialization to the instruction-tuning process. Given the dataset size, the model was further fine-tuned for 5 epochs on non-3D, descriptive property tasks and 1 epoch on 3D property tasks.

**Downstream Tasks: Zero-shot Molecule Editing**. Zero-shot molecule editing was conducted on the curated dataset presented in Liu et al. (2023a) which consists of 200 randomly sampled

molecules from the ZINC dataset. Each molecule was paired with molecule editing prompts (chemical instructions such as *"The molecule is more soluble in water")* and their corresponding SMILES. The dataset included molecular structures that were unseen during training. Starting with the original SMILES, the universal molecular token generated by the trained MQ-Former, and the editing prompt, we generated SMILES of the edited molecule. Using the pretrained MV-CLAM checkpoints from the molecule captioning stage, the model was further fine-tuned for 4 epochs on the PubChem 324k pretraining and training datasets. This fine-tuning enabled MV-CLAM to directly generate SMILES from molecular universal tokens and was crucial to produce valid SMILES, considering the nature of LLaMA's general-purpose tokenizer which was not explicitly trained for SMILES generation.

## A.4 EFFECTIVENESS OF MQ-FORMER

In this appendix section, we demonstrate the efficiency of our proposed MQ-Former, which aligns two modalities (2D and 3D molecular representations) simultaneously to text. We begin by comparing the captions generated by our model to ground truth PubChem label descriptions, showing that our model successfully outputs IUPAC names, generic names, and clinical usages that align with literature references. Next, we visualize the attention maps for our query tokens alongside their distances in latent embeddings, revealing that 2D queries attend to 2D-related textual descriptions and 3D queries to 3D-related ones, with our MQ-Former effectively capturing both modalities. We also compare the captions generated using 2D-only or 3D-only embeddings aligned by the former Q-Former module to highlight the limitations of single-modal approaches. Finally, we examine multi-view embeddings aligned to text by the former Q-Former, demonstrating that simply representing the molecule via multi-view embeddings and aligning them simultaneously does not fully extract diverse perspectives. This analysis emphasizes the need for the novel MQ-Former architecture, which efficiently aligns multi-modal molecular representations to text for richer and more accurate understanding.

### A.4.1 COMPARISON OF MV-CLAM CAPTIONS WITH GROUND TRUTH

We provide caption examples within the test dataset as specified in Section 6.2. MV-CLAM not only correctly generates IUPAC and generic names but also additional information unavailable in ground truth labels.

Table 8: Comparison of ground truth and MV-CLAM descriptions. Matching keywords are highlighted in bold, while additional details provided by MV-CLAM are marked in red.

| Molecule | Ground Truth | MV-CLAM |
|---|---|---|
|  | **Rifapentine** is a **rifamycin antibiotic** that is similar in structure and activity to rifampin and rifabutin and that is used in combination with other agents as **therapy of tuberculosis**, particularly in once or twice weekly regimens. Rifapentine is associated with transient and asymptomatic elevations in serum aminotransferase and is a likely cause of **clinically apparent acute liver injury**. | **Rifapentine** is a **rifamycin antibiotic** that is similar in structure and activity to rifampin and rifabutin and that is used in combination with other agents as **therapy of tuberculosis**, particularly in once or twice weekly regimens. Rifapentine is associated with transient and asymptomatic elevations in serum aminotransferase and is a likely cause of **clinically apparent acute liver injury**. Rifapentine is a **long-acting, cyclopentyl-substituted derivative of rifamycin.** |
|  | **N-(2-hydroxytricosanoyl)-15-methylhexadecasphing-4-enine-1-phosphocholine** is an N-acyl-15-methylhexadecasphing-4-enine-1-phosphocholine in which the **acyl group has 23 carbons and 0 double bonds and is 2-hydroxylated**. It is functionally related to a 15-methylhexadecasphing-4-enine. | **N-(2-hydroxytricosanoyl)-15-methylhexadecasphing-4-enine-1-phosphocholine** is an N-acyl-15-methylhexadecasphing-4-enine-1-phosphocholine in which the **acyl group has 23 carbons and 0 double bonds and is 2-hydroxylated**. It is functionally related to a 15-methylhexadecasphing-4-enine. |

### A.4.2 ATTENTION MAP VISUALIZATION

We provide the images of attention map explained in Section 6.3 (Appendix Figures 5, 6). Attention map of the shared self-attention layers is visualized to compare the processing of 2D and 3D query tokens. As shown in the figures, the query tokens for each dimension exhibit distinct attention

patterns across the sentence. To further analyze the embeddings of 2D, 3D queries, and our universal query tokens, we visualized them in the latent space alongside the word embeddings of "water" - a chemical property with high attention to 2D - and "3"-a positional coordinate with high attention to 3D (Appendix Figure 7). The results reveal that the universal query token maintains moderate distances to both word embeddings, reflecting the interplay between 2D and 3D molecular views. This demonstrates that MQ-Former effectively preserves modality-specific information from 2D and 3D while aligning seamlessly with textual semantics.

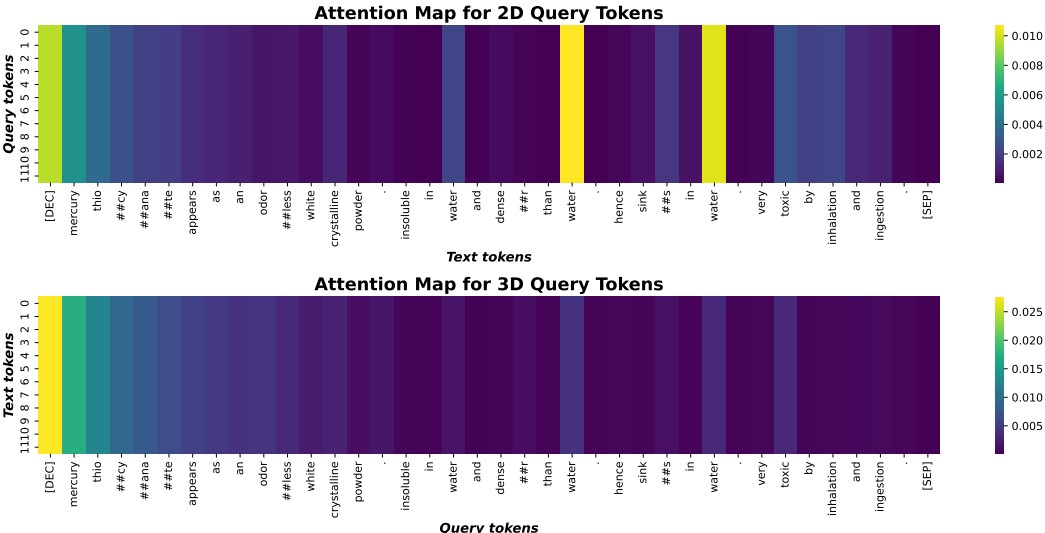

Figure 5: Attention map visualization. 2D query tokens focus on chemical properties like water solubility present in text descriptions.

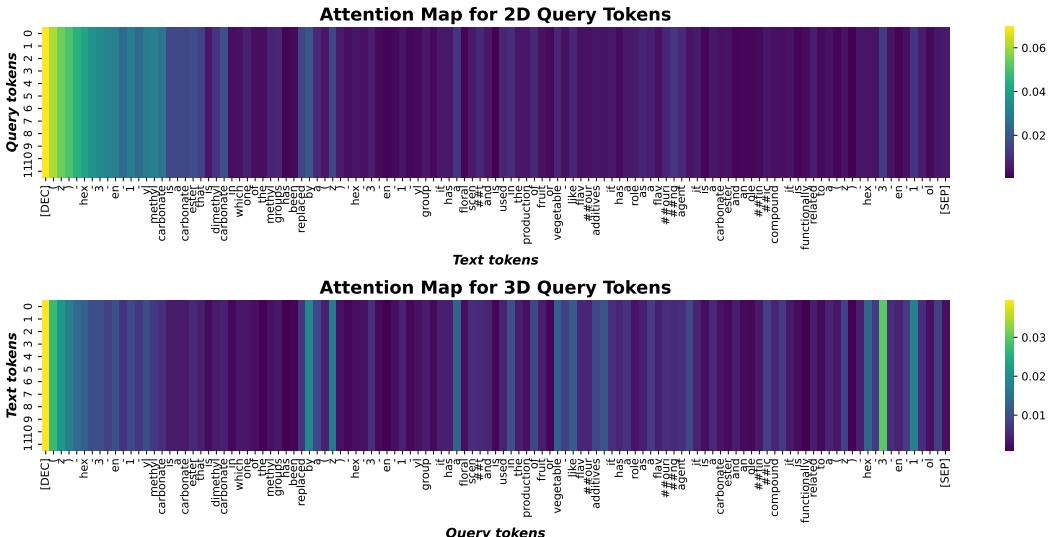

Figure 6: Attention map visualization. 3D query token focuses on positional information of atoms in text descriptions.

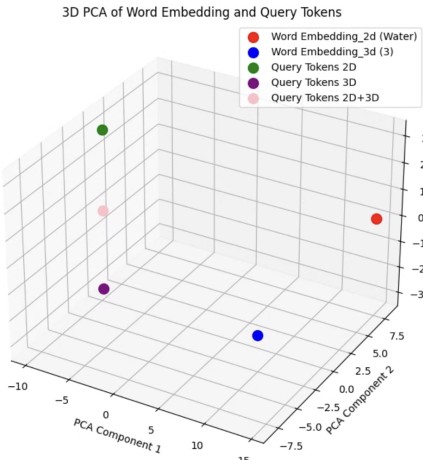

Figure 7: Latent space representation of query tokens and word embeddings, illustrating the alignment of 2D, 3D, and universal queries with textual semantics.

### A.4.3 SINGLE-MODALITY CAPTION ALIGNMENT

Appendix Figure 8 highlights the differences in captioning results between the uni-modal Q-Former ablation models and our multi-view approach. This demonstrates that the multi-view approach generates richer and more precise molecular descriptions as mentioned in Section 6.3.

| 2D Only | 3D Only | Original | Ground Truth |
|---|---|---|---|
| **isatinic acid** is a member of the class of 4-aminobenzoic acids that is anthranilic acid substituted by a hydroxy group at C-5. It has a role as a bacterial metabolite. It is a monohydroxybenzoic acid and a member of 4-aminobenzoic acids. It is functionally related to an anthranilic acid. It is a conjugate acid of an isatinate. Anthraniloic acid is a metabolite found in or produced by Escherichia coli (stra | **4-hydroxyphenyl sulfate(1-)** is a phenyl sulfate oxoanion that is the conjugate base of 4-hydroxyphenyl hydrogen sulfate, obtained by deprotonation of the sulfate group; major species at pH 7. 3. It has a role as a human metabolite. It is a conjugate base of a 4-hydroxyphenyl hydrogen sulfate. Phenyl hydrogen sulfate is a metabolite found in or produced by Escherichia col | **(R)-3-hydroxytriacontanoyl-CoA** is a 3-hydroxy fatty acyl-CoA that results from the formal condensation of the thiol group of coenzyme A with the carboxy group of (R)-3-hydroxytriacontanoic acid. It is a (R)-3-hydroxyacyl-CoA, a 3-hydroxy fatty acyl-CoA and an ultra-long-chain fatty acyl-CoA. It is a conjugate acid | **(R)-3-hydroxytriacontanoyl-CoA** is a 3-hydroxy fatty acyl-CoA that results from the formal condensation of the thiol group of coenzyme A with the carboxy group of (R)-3-hydroxytriacontanoic acid [(R)-3-hydroxymelissic acid]. It is a (R)-3-hydroxyacyl-CoA, a 3-hydroxy fatty acyl-CoA and an ultra-long-chain fatty acyl-CoA. It is functionally related to a triacontanoic acid. It is a conjugate acid of a (R)-3-hydroxytriacontanoyl-CoA(4-) |
| 2D only | 3D only | Original | Ground Truth |

Figure 8: Comparison of Uni-modal Q-Former Ablation and Ours

### A.4.4 MULTI-VIEW REPRESENTATION ANALYSIS

To highlight the necessity of MQ-Former, we conducted an ablation study comparing our architecture with a variant that aligns multi-view molecular representations using a single Q-Former module. The multi-view molecular embedding was constructed by concatenating the 2D embeddings from MAT and the 3D embeddings from Uni-Mol, then projected to textual space using the Q-Former. Unlike the concatenation-based approach, MQ-Former preserves the rich, distinct representations of molecular views. This design facilitates more fine-grained alignment with text, maintaining di-

versified information, which results in higher-quality captions across all evaluated metrics (Table 9). Overall, MQ-Former enables the preservation of detailed and diverse molecular representations, facilitating precise alignment with textual descriptions and delivering superior performance in the captioning task.

Table 9: Captioning Performance Comparison: Multi-View Representation with Single Q-Former

| Model | BLEU-2 | BLEU-4 | ROUGE-1 | ROUGE-2 | ROUGE-L | METEOR |
|---|---|---|---|---|---|---|
| Multi-view + Q-Former | 29.80 | 22.70 | 39.07 | 24.92 | 33.09 | 35.49 |
| MV-CLAM | 31.75 | 24.48 | 40.43 | 25.72 | 33.79 | 36.54 |

## A.5  ZERO-SHOT MOLECULE EDITING

We provide more examples of successful zero-shot molecule editing cases given chemical property based instructions (Appendix Figure 9,10,11,12). The values presented indicate the predicted LogP (octanol-water partition coefficient), topological surface area (TPSA), quantitative estimate of drug-likeness (QED) and number of hydrogen bond and acceptors. Each figure showcases original molecules alongside their modified counterparts with numerical indicators representing the chemical properties before and after the zero-shot editing. LogP values reflect solubility in water, while topological surface area relates to molecular permeability. QED reflects drug likeliness. The modifications are aligned with targeted property-based editing prompt, demonstrating the flexibility and chemical expertise of MV-CLAM.

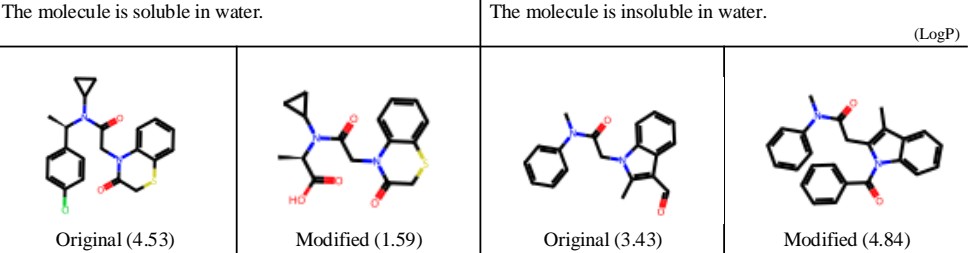

Figure 9: Editing Solubility (LogP Adjustments): Smaller LogP indicates higher solubility in water. Molecules were successfully modified given the prompt *"The molecule is soluble/insoluble in water"*.

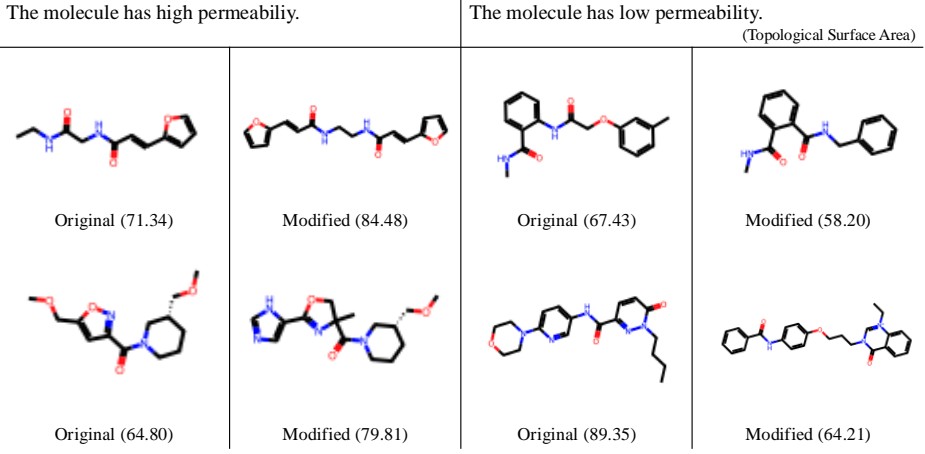

Figure 10: Editing Permeability (Topological Surface Area, TPSA Adjustments): A higher TPSA implies lower permeability, while a lower TPSA suggests higher permeability. Molecules were successfully modified given the prompt *"The molecule has high/low permeability"*.

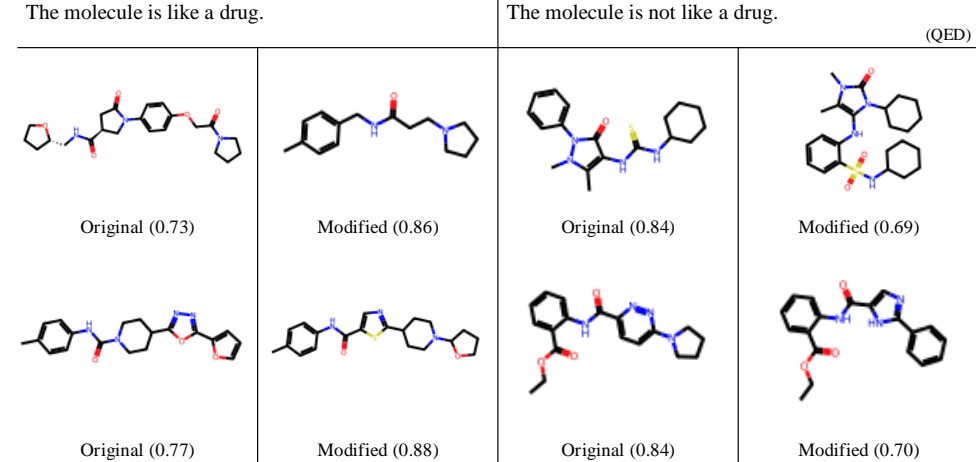

Figure 11: Editing Drug Likeliness (Quantitative Estimate of Drug-likeness, QED): A higher QED suggests a compound is more likely to possess favorable pharmacokinetic and ADMET (absorption, distribution, metabolism, excretion, and toxicity) properties, being more drug-likely. Molecules were successfully modified given the prompt *"The molecule is/is not like a drug"*.

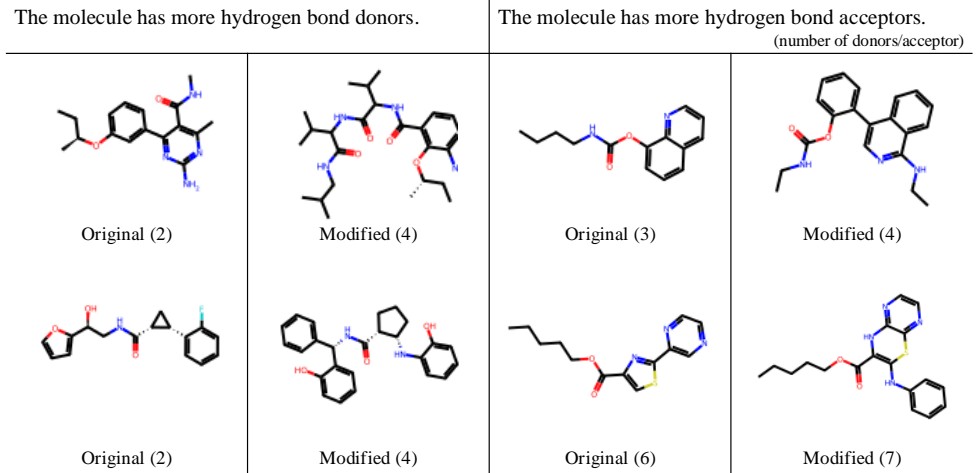

Figure 12: Editing Hydrogen Bond Acceptor/Donors: The number of hydrogen bond acceptors and donors in the molecule were given for evaluation. Molecules were successfully modified given the prompt *"The molecule has more hydrogen bond donors/acceptors"*.

A.6  ABLATION STUDIES FOR STAGE 1. TRAINING MQ-FORMER

To better understand the contributions of individual components in our model, we conducted a series of ablation studies focusing on three factors: the graph encoder architecture, the training loss design, number of query tokens used in the model. We report the preliminary results retrieval metrics for the first stage of pretraining MQ-Former. Although early molecule-text retrieval results do not directly translate to improved molecule captioning outcomes, they have a tendency to exhibit positive correlation in previous studies.

**Graph Encoder Ablation**   We examine three variations of 2D graph encoders, all of which remain frozen during MQ-Former training (Appendix Table 10). Under a consistent 3D encoder configuration, we report retrieval metrics for GIN initialized randomly, MAT embeddings adjusted via an additional linear layer for size reduction, and preserved MAT embeddings. The results illustrate that the quality of graph encoders significantly influenced the initial performance during the first

stage of pretraining MQ-Former. This observation was a key motivation behind MQ-Former; maintaining high-quality embeddings from pretrained graph encoders appears to be effective for textual alignment.

Table 10: Retrieval performance comparison in batch and test set for different 2D graph encoders.

| Model | Retrieval in batch | | | | Retrieval in test set | | | |
| | M2T | | T2M | | M2T | | T2M | |
| | ACC | R@20 | ACC | R@20 | ACC | R@20 | ACC | R@20 |
|---|---|---|---|---|---|---|---|---|
| Random | 87.42 | 99.54 | 87.31 | 99.54 | 38.87 | 88.59 | 37.54 | 88.03 |
| MAT_linear | 90.38 | 99.64 | 89.26 | 99.64 | 55.96 | 90.84 | 54.37 | 90.69 |
| Ours | **96.16** | **99.85** | **96.06** | **99.85** | **67.72** | **96.62** | **68.69** | **95.86** |

**Number of Query Tokens** We conducted a preliminary ablation study comparing the use of a single query token versus multiple query tokens (Appendix Table 11). We also showcase an attention map (Appendix Figure 13) to show multiple query tokens allow the model to capture distinct attention patterns in textual descriptions. This decision aligns with the design philosophy of BLIP-2 (Li et al., 2023) and ensures that MQ-Former is capable of leveraging the unique information provided by each modality for more comprehensive molecule captioning.

Table 11: Retrieval performance comparison in batch and test set for different number of query tokens.

| Model | Retrieval in batch | | | | Retrieval in test set | | | |
| | M2T | | T2M | | M2T | | T2M | |
| | ACC | R@20 | ACC | R@20 | ACC | R@20 | ACC | R@20 |
|---|---|---|---|---|---|---|---|---|
| 1 Query Token | 96.16 | 99.85 | 95.40 | 99.85 | 70.08 | 96.42 | 70.97 | 95.5 |
| 12 Query Tokens | 96.73 | 99.90 | 96.01 | 99.85 | 70.90 | 96.98 | 71.15 | 95.96 |

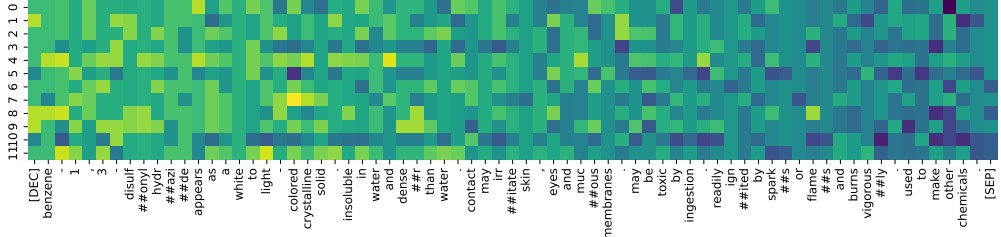

Figure 13: Attention map of length 12 molecular query token. Different queries attend to different words within the textual descriptions, allowing comprehensive alignment between molecules and text.

**Training Loss Ablation** We also evaluated the effect of loss weighting in the multi-objective training framework, along with the evaluation of symmetric components in molecule-text contrasting loss (Appendix Table 12). These findings demonstrate that amplifying the LM loss weight better aligns molecular and textual representations, justifying its use in subsequent training stages. Due to different batches within experiments, we only report the metrics for the entire test set.

## A.7 ABLATION STUDIES FOR STAGE 2. SPECIALIZING LLAMA2 FOR MOLECULE CAPTIONING

**1D Molecular Representations** We conducted an ablation study to compare the use of SELFIES (Krenn et al., 2020) with SMILES as input representations (Appendix Table 13). Using the pretrained Stage 2 checkpoint, the model was further trained for captioning under identical settings. After 10 stages of training with SELFIES, SMILES consistently demonstrated superior performance across metrics such as BLEU, METEOR, and ROUGE, validating the effectiveness of our selection.

Table 12: Retrieval performance comparison in test set for training loss weight and components.

| Model | M2T | | T2M | |
|---|---|---|---|---|
| | ACC | R@20 | ACC | R@20 |
| lm loss * 1 | 69.87 | 97.75 | 69.26 | 95.55 |
| Ours | 70.90 | 96.98 | 71.15 | 95.96 |

Table 13: Captioning performance comparison for 1D molecular representations

| Model | BLEU-2 | BLEU-4 | ROUGE-1 | ROUGE-2 | ROUGE-L | METEOR |
|---|---|---|---|---|---|---|
| SELFIES | 28.39 | 20.89 | 33.25 | 37.58 | 22.49 | 31.37 |
| SMILES | 31.75 | 24.48 | 40.43 | 25.72 | 33.79 | 36.54 |

## A.8 FAILURE CASE STUDY

Appendix Table 14 showcases two instances where MV-CLAM fails to differentiate structurally similar molecules. First, the model misclassifies lactoyl-CoA as oleoyl-CoA despite the key difference being the length of the carbon chain. This indicates a limitation in the model's capacity to capture subtle variations in carbon chain lengths. Second, the model misidentifies Ajugaciliatin B as subtypes E and C, demonstrating that while it successfully recognizes the molecule's primary backbone, it struggles to distinguish the small functional groups that define each subtype. This suggests that the model is not sufficiently sensitive to minor structural modifications. Both errors appear to stem from the model's difficulty in perceiving refine differences in chemical properties and spatial structure between the ground truth and its predictions. This underscores a broader challenge in molecular captioning: capturing subtle yet critical molecular features that may not greatly impact the primary structure but are crucial contributors for property.

To overcome these limitations, we propose several future studies. First, expanding our MQ-Former to align additional views or modalities, along with finer-grained molecular or related biological embeddings, could offer complementary insights to enhance the model's ability to differentiate between similar molecules. This multi-view alignment could offer a more holistic understanding of the molecule's structure and properties. In addition, curating larger molecule datasets would enhance the model's capacity to generalize, ensuring it has sufficient exposure to a wide range of molecular variations during training. These developments will address the current shortcomings and pave the way for more accurate molecular identification in future iterations of the model.

Table 14: Failure Case Study: Comparison of Ground Truth and MV-CLAM Descriptions

| Molecule | Ground Truth | MV-CLAM |
|---|---|---|
|  | **Lactoyl-CoA(4-)** is an acyl-CoA(4-) that is the tetraanion of lactoyl-CoA arising from deprotonation of phosphate and diphosphate functions. It is a conjugate base of a lactoyl-CoA. | **oleoyl-CoA(4-)** is an acyl-CoA(4-) that is the tetraanion of oleoyl-CoA, arising from deprotonation of phosphate and diphosphate functions. It has a role as a human metabolite and a Saccharomyces cerevisiae metabolite. It is a saturated fatty acyl-CoA(4-) and a 3-substituted propionyl-CoA(4-). It is a conjugate base of an oleoyl-CoA |
|  | **Ajugaciliatin B** is a diterpene lactone isolated from the whole plants of Ajuga ciliata and has been shown to exhibit neuroprotective activity. It has a role as a plant metabolite and a neuroprotective agent. It is a diterpene lactone, an acetate ester, a butenolide, a carbobicyclic compound, an organochlorine compound and a tertiary alcohol. Ajugaciliatin B is a natural product found in Ajuga ciliata with data available. | **ajugaciliatin C** is a natural product found in Ajuga ciliata with data available. **Ajugaciliatin E** is a diterpene lactone isolated from the whole plants of Ajuga ciliata. It has a role as a plant metabolite. It is a butenolide, an acetate ester, a diterpene lactone and a organochlorine compound. It is functionally related to a tiglic acid. **Ajugaciliatin E** is a natural product found in Ajuga ciliata |

