# OpenReview forum: "MV-CLAM: Multi-View Molecular Interpretation with Cross-Modal Projection via Language Model"
_ICLR.cc/2025/Conference — Submitted to ICLR 2025_

### Official Review · Reviewer_R1Xc · 2024-10-30

**Soundness:** 2
**Presentation:** 2
**Contribution:** 2
**Rating:** 3
**Confidence:** 4

**Summary:**

The paper proposes MQ-Former, an extension of the Q-Former framework, which incorporates a multi-query mechanism for aligning both 2D and 3D molecular data with textual information for enhanced molecule-text retrieval and molecule captioning.

**Strengths:**

- The paper aims to enhance cross-modal alignment by integrating 2D and 3D molecular views.
- The model demonstrates improvements in molecule-text retrieval and captioning performance over baseline models.
- The paper includes case studies and examples of zero-shot molecule editing.

**Weaknesses:**

- The model lacks significant innovation, as MQ-Former primarily adds an extra branch to the existing Q-Former with only minor variations in training objectives.
- Experiments are restricted to molecule-text retrieval and captioning on PubChem. The paper lacks essential molecular tasks like molecule generation and datasets like ChEBI-20.
- The motivation for adding a branch to Q-Former, rather than simply using a 3D molecular encoder like prior works (e.g., 3D-MoLM), is unclear.
- The paper’s presentation could be improved. Plots lack careful formatting, with text that is difficult to read due to small font sizes.

**Questions:**

- How does MQ-Former handle scenarios where 2D and 3D molecular information may not equally contribute to textual descriptions?
- Could the authors include more molecular tasks, such as molecule generation or property prediction, to provide a more comprehensive evaluation of MQ-Former?
- What impact does the weighting of the multi-objective training loss have on the model’s performance?

---

> ### Author Response · Authors · 2024-11-24
> **Response to Reviewer R1Xc's comments - 1**
>
> **Substantial support on novelty and technical contributions**
> > “How does MQ-Former handle scenarios where 2D and 3D molecular information may not equally contribute to textual descriptions?”
>
> Thank you for this insightful question, which directly aligns with the motivation behind our research. We believe that 2D and 3D molecular representations capture distinct and complementary aspects of molecular information, particularly in their connection to textual descriptions. For example, 2D information encodes connectivity and topological features, while 3D information represents spatial and geometric properties that are critical for describing stereochemistry or binding conformations. As demonstrated in Appendix Figure 5-6, 2D and 3D query tokens obtained from our MQ-Former architecture attend to different words in the textual descriptions. This highlights that each modality contributes uniquely to understanding the molecular-text relationship.
>
> To address this, our renovated MQ-Former architecture aligns 2D and 3D multi-view embeddings simultaneously to the text. This simultaneous alignment ensures that both modalities are leveraged in a complementary manner, allowing the model to capture richer molecular semantics and better connect them with textual descriptions. We further validate the novelty and technical constructions of our approach using embedding space visualization and additional ablation study. Please kindly refer to “Concerning novelty and technical contributions” in General Response for details.
>
>
> **Construct more downstream tasks**
>
> **1. QA results unavailable in main text**: Thank you for your valuable comment. We have evaluated the QA experiment results by comparing our framework, MV-CLAM—integrating the Multi-querying Transformer module with both 3D and 2D molecular encoders—against frameworks that use a single molecular encoder. Specifically, we compared it to:
>
> - 3D-MoLM: Q-Former with a 3D molecular encoder
> - 2D-MoLM: Q-Former with a 2D molecular encoder
>
> For each case, including our framework, we reproduced QA results for both non-3D properties (Molecular Weight, LogP, Complexity, and Topological Polar Surface Area) and 3D properties (HOMO, LUMO, HOMO-LUMO gap, and SCF energy), as shown in Table R5 and R6.
>
> **Table R5.** Comparison of QA results for non-3D molecular property prediction. The values in parentheses indicate the validity score.
> | Model            | Molecular Weight | LogP  | Complexity | Topological Polar Surface Area |
> |-------------------|-----------------------|------------|------------------|-------------------------------------|
> | 2D-MoLM          | 47.51 (0.98)         | 0.89 (0.99)| 110.78 (0.99)    | 16.65 (0.99)                        |
> | 3D-MoLM          | 42.76 (0.96)         | 1.25 (0.96)| 105.03 (0.96)    | 20.97 (0.92)                        |
> | MQ-Former (Ours) | **21.35 (0.92)**     | **0.69 (0.94)** | **55.14 (0.91)** | **9.65 (0.91)**                     |
>
>
> **Table R6.** Comparison of QA results for 3D molecular property prediction. The values in parentheses indicate the validity score.
> | Model            | HOMO | LUMO  | HOMO-LUMO | SCF Energy |
> |-------------------|------------|------------|----------------|-----------------|
> | 2D-MoLM          | 0.78 (0.99)| 0.47 (0.99)| 0.39 (0.90)    | 0.98 (1.00)     |
> | 3D-MoLM          | 0.42 (0.99)| 0.44 (0.98)| 1.26 (0.99)    | 1.22 (0.98)     |
> | MQ-Former (Ours) | **0.35 (0.98)** | **0.42 (0.93)** | **0.35 (0.99)** | **0.32 (0.99)** |
>
>
> Bold text indicates the best performance. As shown in the tables, our framework with MQ-Former achieved the highest scores in both non-3D and 3D molecular property QA tasks by effectively utilizing molecular information from both dimensions. We have now properly updated the result tables as Table 5 in the revised manuscript.

---

> > ### Author Response · Authors · 2024-11-24
> > **Response to Reviewer R1Xc's comments - 2**
> >
> > **2. Molecule generation: LLaMA2 based models:** Thank you for the thoughtful feedback. We chose LLaMA2 as the base large language model due to its strong performance in molecule-text modeling, as demonstrated in related works like 3D-MoLM and UniMoT. LLaMA2’s ability to leverage a large academic corpus aligns with our goal of developing a refined chemical foundation model. While MQ-Former is model-agnostic and can work with other architectures (e.g., T5-based models), we prioritized consistency with prior studies for fair comparisons, emphasizing the performance of our novel cross-modal projector.
> >
> > However, an important limitation of using LLaMA2 is that its tokenizer is not optimized for generating chemical SMILES representations. This limitation arises because LLaMA tokenizers are pretrained on general-purpose and academic text corpora, which do not include specialized tokenization for chemical structures like SMILES. As a result, our primary downstream task focuses on molecule captioning and molecule-text alignment rather than molecule generation. Previous models based on LLaMA (e.g., 3D-MoLM) also do not focus on SMILES generation without additional tokenizer modifications or pretraining specific to chemical domains.
> >
> > While the LLaMA2 tokenizer is not explicitly trained to generate chemical SMILES representations, we sought to demonstrate the flexibility and potential of our MQ-Former architecture by exploring **zero-shot molecule editing** as an additional capability. In our experiments, we showcased instances where the model successfully edited molecule SMILES in a zero-shot setting, highlighting the generalization ability of MV-CLAM and its capacity to handle SMILES generation through the raw LLaMA2 tokenizer without further specialization. This approach represents, to the best of our knowledge, the first attempt to generate and edit molecular SMILES using the unmodified LLaMA tokenizer, directly bridging the gap between chemical structure representations and pretrained large language models. We have revised Section 6.5 and Appendix A.3 in the manuscript to incorporate these insights.

---

> > > ### Author Response · Authors · 2024-11-24
> > > **Response to Reviewer R1Xc's comments - 3**
> > >
> > > **Image/Formatting**: Thank you for your feedback regarding the presentation of the paper. We have carefully revised the plots to enhance their readability by adjusting the formatting and increasing font sizes for clarity. Especially for the image in Appendix A.6. Zero Shot Molecule Editing, we have separated the image to Appendix Figures 9~12 with captions for clarity.
> > >
> > > **Training loss weight ablation**: We truly value your detailed comments. To assess the impact of the weighting in the multi-objective training loss, we conducted experiments validating its effect on Stage 1 metrics, specifically molecule-text retrieval performance on the pretraining dataset. These evaluations were conducted at epoch 10. Based on the preliminary results observed at epoch 10, we analyzed the impact of loss weighting on molecule-text retrieval metrics (M2T and T2M), as shown in Table R9. The table indicates a clear tendency: amplifying the language model (LM) loss weight by a factor of 2 improves both accuracy (ACC) and recall at rank 20 (R@20) across the evaluated tasks. Specifically, we observed that the ACC for M2T increased from 69.87 to 70.90, and for T2M, from 69.26 to 71.15. Although there was a slight decrease in R@20 for M2T (from 97.75 to 96.98), T2M R@20 remained stable.
> > >
> > > These preliminary findings suggest that amplifying the LM weight helps to better align the molecular and textual representations, leading to improved performance. Consequently, we adopted this adjusted weighting strategy for the subsequent stages of our experiments. We have organized additional ablation studies and its results in Appendix A.7.
> > >
> > > **Table R9.** Ablation Study: Training loss weights
> > > | Model | M2T ACC | M2T R@20 | T2M ACC | T2M R@20 |
> > > |-------|---------|----------|---------|----------|
> > > | Lm*1  | 69.87   | 97.75    | 69.26   | 95.55    |
> > > | Lm*2  | 70.90   | 96.98    | 71.15   | 95.96    |

---

### Official Review · Reviewer_48Rx · 2024-10-30

**Soundness:** 3
**Presentation:** 2
**Contribution:** 2
**Rating:** 5
**Confidence:** 3

**Summary:**

The paper introduces MV-CLAM, a framework utilizing a novel multi-querying transformer (MQ-Former) to enhance the alignment of multi-modal molecular representations with text. By employing a shared self-attention layer, this approach effectively consolidates 2D and 3D molecular data into query tokens, improving performance in molecule-text retrieval and captioning tasks. Additionally, it demonstrates potential for zero-shot molecule editing and molecule-related question answering, thereby facilitating better characterization of chemical structures.

**Strengths:**

* The description of the proposed methodology is easy to follow. The paper is well written in general.
* The paper introduces a promising multi-view for approach for the infusion of specialized chemical knowledge into general-purpose pre-trained LLMs.
* The proposed MV-CLAM achieves state-of-the-art on PubChem324K for molecule captioning and retrieval tasks.

**Weaknesses:**

* The experimental evaluation of the proposed method is conducted on a single dataset for both task: molecule captioning and molecule-text retrieval.
* The list of baseline models on molecule captioning only includes a single T5 language model while there are more recent works, including: nach0 and Text+ChemT5.
* Some implementation decisions are not justified well enough. This includes: (i) the choice of SciBERT as a language encoder for MQ-Former; (ii) the choice of 2D and 3D encoders; (iii) introduction of $K$ query tokens instead of a single query token for each view; (iv) the choice of LLaMA2 as an LLM. It is unclear how the experimental results would change if each of the mentioned models is replaced with another one.
* Incomplete ablation study. The necessity of (i) Molecule-text Contrasting and (ii) Molecule-text Matching losses is not proven experimentally. For (i), it is unclear whether two loss components required or the model will perform well with a single one. For (ii), the impact of negative sample is under-explored.
* The effect of most hyper-parameters in the method's module on the resulting performance is understudied. For instance, query token count, negative sample count in MTM loss.
* The methodology for molecule-text retrieval is unclear from the paper.
* The applicability of the proposed methodology to broader list of datasets is questionable: it requires 2D/3D molecular data in addition to simple SMILES string representations.

**Questions:**

* Add experimental comparison against more chemical language models on molecule captioning, e.g., nach0 [1], Text+Chem T5 [2], SciFive [3], PRESTO [4], GitMol [5].
* For retrieval task (Table 1), is it possible to add chemical BERT-based encoders in addition to textual encoder SciBERT? (e.g., ChemBERTa)
* Conduct additional experiments on other molecule captioning datasets such as Mol-Instructions [6] and CheBI20 [7].
* For molecule-text retrieval, do you adopt a generative approach (e.g., GENRE [8]) or the task is formulated as a cross-modal embedding-based search by similarity (e.g., as in [9])?
* In Figure 3, where does the textual description come from during prediction on a test set? As far as I understand the molecule captioning task, you are only given a SMILES string.
* What is the LLaMA version you use? Add adopted HuggingFace checkpoints.
* Even if you adopt a LLaMA with 7B parameters, MolT5 has less than 1B. Could not we just scale MolT5 to 3-5B parameters and obtain a better molecule captioning quality?
* Why is MolT5 absent from the Table 1?
* Add ablation study for SciBERT, 2D/3D molecule encoders, LLaMA2.
* Add ablation study for training losses. For Molecule-text Contrasting loss, prove it requires two components. For Molecule-text Matching loss, explore the effect of negative samples.
* Is it possible to generalize the methodology to unseen datasets and unseen SMILES? Given a SMILES, can I always obtain its 2D/3D representation and apply a pre-trained MV-CLAM model?




Typos:
* Line 102: transformer -> Transformer, Add reference.
* Line 194: **$A$** under-specified.
* Line 234: Missing citation for LoRA.

---

> ### Author Response · Authors · 2024-11-24
> **Response to Reviewer 48Rx's comments - 1**
>
> **Baseline models and datasets**: Please kindly refer to *General Response “ (1). Concerning SOTA baseline models” and “(2). Concerning the CheBI20 dataset*”.
>
> **Adding additional baseline models (ChemBERTa, MolT5) for the retrieval task (Table 1)**:
>
> Thank you for the insightful comment. ChemBERTa, as a chemical BERT-based encoder, could be considered as an alternative to SciBERT. However, unlike SciBERT, ChemBERTa is pretrained specifically on SMILES strings derived from chemical databases like PubChem and ChEMBL, excelling in tokenizing SMILES using a Byte-Pair Encoding (BPE) tokenizer. Given the MQ-Former architecture, which takes both molecular graphical structures and textual descriptions as inputs, we utilized the architecture and pretrained weights of SciBERT, pretrained on a large corpus of scientific texts. Additionally, we adhered to approaches from *prior molecule-captioning research using Q-Former, such as 3D-MoLM and UnimoT, which demonstrated the effectiveness of SciBERT in similar tasks*.
>
> MolT5, based on the T5 (Text-to-Text Transfer Transformer) architecture, is optimized for sequence generation, making it less efficient for molecule-text retrieval tasks compared to bidirectional encoders like SciBERT, which are better suited for retrieval. Its training scheme also differs from ours and the selected baseline models. Consequently, *prior works (3D-MoLM, UnimoT, MolCA) have not included MolT5 in molecule-text retrieval comparisons*, focusing instead on molecule-captioning tasks, as we do.

---

> ### Author Response · Authors · 2024-11-24
> **Response to Reviewer 48Rx's comments - 2**
>
> **Justification for implementation decisions**
>
> Thank you for your insightful comments. We appreciate the opportunity to clarify the implementation decisions and their rationale. Below, we address each of the points raised and provide additional justifications:
>
> 1. **Choice of SciBERT as the Language Encoder for MQ-Former**
> > The MQ-Former module is initialized using SciBERT checkpoints because SciBERT has been pre trained on scientific texts, including chemical literature, and is well-suited for extracting textual features from domain-specific datasets. This aligns with the initialization strategy employed by related works such as MolCA and 3d-MoLM. For molecule captioning, the text embeddings are subsequently encoded and decoded using the LLaMA2 tokenizer, enabling us to combine SciBERT’s scientific domain expertise with LLaMA2’s strong language modeling capabilities. Our intention was to keep the encoders consistent with prior works to establish a fair baseline while focusing on evaluating the cross-modal projection capabilities of MQ-Former.
> > To improve clarity in the manuscript, we have rephrased sections to better explain this dual-encoder strategy. Specifically, the "text encoder" subsection has been removed, and the details have been integrated into Section 3.2 (MQ-Former) and Appendix A.3 (Experimental Settings) to provide a more cohesive explanation.
>
> 2. **Choice of 2D and 3D Encoders**
> > We adopted state-of-the-art graph-based molecular encoders for 2D (Molecular Attention Transformer) and 3D (Uni-Mol). These choices are motivated by their proven effectiveness in molecular representation learning and their alignment with the architecture of 3D-MoLM. While we agree that exploring alternative 2D and 3D encoders is valuable, our primary goal in this study was to demonstrate MQ-Former’s cross-modal projection advancements when aligning 2D embeddings simultaneously with 3D embeddings. Future work could replace these encoders with alternatives to further generalize the approach.
>
> 3. **Introduction of Query Tokens Instead of a Single Query Token**
> > The decision to utilize multiple learnable query tokens in our MQ-Former architecture is inspired by the original work in BLIP-2, which also employs multiple learnable query tokens for cross-modal alignment. This approach has been shown to enhance the model's capacity to attend to diverse aspects of visual or structural representations and align them effectively with textual descriptions. Previous works (MolCA, 3D-MoLM) also use multiple tokens instead of one.
> > We also conducted a preliminary ablation study comparing the use of a single query token versus multiple query tokens affecting molecule-text retrieval performance on the pretraining dataset (Table R8). These evaluations were conducted at epoch 10. We also showcase an attention map (Appendix Figure 13) to show multiple query tokens allow the model to capture distinct attention patterns in textual descriptions. This decision aligns with the design philosophy of BLIP-2 [15] and ensures that MQ-Former is capable of leveraging the unique information provided by each modality for more comprehensive molecule captioning. The results have been organized into Appendix A6 (Number of Query Tokens).
>
> **Table R8**.  Ablation study: Number of query tokens
>
> | Inbatch | M2T       |         | T2M       |         | Full | M2T       |         | T2M       |         |
> |---------|-----------|---------|-----------|---------|------|-----------|---------|-----------|---------|
> |         | ACC       | R@20    | ACC       | R@20    | ACC  | R@20      | ACC     | R@20      |         |
> | 1       | 96.16     | 99.85   | 95.40     | **99.85**   | 70.08| 96.42     | 70.97   | 95.50     |         |
> | 12      | **96.73**     | **99.90**   | **96.01**     | **99.85**   | **70.90** | **96.98**     | **71.15**   | **95.96**     |         |
>
> 4. **Choice of LLaMA2 as the Base Language Model**
> > We chose LLaMA2 as the base large language model due to its strong performance in molecule-text modeling, as demonstrated in related works like 3D-MoLM and UniMoT. By utilizing LoRA, the actual parameters used for training is only 0.29% of the total parameters in the LLaMA2-7B, comparable even to the size of small MolT5-1B. While MQ-Former is model-agnostic and can work with other architectures (e.g., T5-based models), we prioritized *consistency with prior studies (3D-MoLM, UniMoT)* for fair comparisons, emphasizing the performance of our novel cross-modal projector, MQ-Former.
>
> References
> > [15] Li et al., "BLIP-2: Bootstrapping Language-Image Pre-training with Frozen Image Encoders and Large Language Models"

---

> ### Author Response · Authors · 2024-11-24
> **Response to Reviewer 48Rx's comments - 3**
>
> **Ablation Studies for MQ-Former training loss**
>
> In cross-modal contrastive learning frameworks like CLIP [16], symmetric loss functions are used to calculate both image-to-text (i→t) and text-to-image (t→i) contrastive losses. This ensures that both modalities (in our case, molecular and textual) are equally optimized for alignment in the shared embedding space.
> For our MQ-Former framework, we adopt a similar principle, calculating molecule-to-text (mol→text) and text-to-molecule (text→mol) contrastive losses. These are integral components in molecular-text modeling tasks, as they ensure bidirectional alignment between molecular representations and textual descriptions. The total symmetric contrastive loss written in *Equation 4* ensures bidirectional alignment: encouraging molecular representation to match its corresponding text representation while contrasting with other text representations in the batch and vice versa. Together, these components enhance the MQ-Former’s ability to create robust molecular-text alignments, leveraging both 2D and 3D molecular structures in a shared embedding space.
> We plan to delve deeper into the exploration of negative samples in future experiments. Thank you for your valuable insight.
>
> **Applicability to unseen datasets**
>
> We acknowledge the concern about the broader applicability of our methodology. Our current dataset only includes SMILES strings and corresponding text descriptions - we preprocess the data to autonomously generate the necessary 2D molecular graphs and 3D conformers based on the SMILES representation using RDKit. This ensures scalability to any dataset containing SMILES strings.
> Additionally, the purpose of using MQ-Former is to leverage pretrained molecular encoders, which are specifically designed to handle 2D and 3D molecular representations. Molecular embeddings to be given as input to MQ-Former is generated automatically using the pretrained encoders. Hence, we can process unseen SMILES strings to generate the required molecular representations (2D graphs and 3D conformers) and subsequently output captions. Thus, while our approach requires intermediate molecular representations, it is fully compatible with datasets as long as they provide SMILES strings.
>
> **Explanation on the methodology for molecule-text retrieval**
>
> For molecule-text retrieval, our approach is formulated as a cross-modal embedding-based search by similarity. As shown in our implementation, we calculate similarity scores between graph (molecular) embeddings and text embeddings using matrix multiplication, rank items based on these scores, and evaluate performance using ranking metrics such as accuracy and recall. This approach aligns with methods that leverage embedding similarity for retrieval, rather than a generative framework like GENRE.
>
> **In Figure 3, where does the textual description come from during prediction on a test set? As far as I understand the molecule captioning task, you are only given a SMILES string.**
>
> For clarification, the PubChem324k test dataset is composed of SMILES-text description pairs. MQ-Former generates universal molecular tokens (molecule SMILES → 2D/3D pretrained embeddings → universal molecular token) to provide as input for generating molecular captions via LlaMA2. The generated captions are compared with ground truth description labels using the metrics BLEU, METEOR and ROUGE.
>
> **What is the LLaMA version you use? Add adopted HuggingFace checkpoints.**
>
> We used the following version of LLaMA2 in huggingface, compatible with Transformers. baffo32/decapoda-research-llama-7B-hf (https://huggingface.co/baffo32/decapoda-research-llama-7B-hf). We have clarified this in our manuscript Appendix A.3. as well. Thank you.
>
> **Typos and missing references**
>
> Thank you for pointing out the typos and missing references. We appreciate your attention to detail, and have carefully revised the manuscript to correct the typos and ensure that all references are properly cited.
>
> References
> > [16] Radford et al., "Learning Transferable Visual Models From Natural Language Supervision"

---

> > ### Comment · Reviewer_48Rx · 2024-12-01
> > **Reviewer Response**
> >
> > Dear authors,
> >
> > I have read all your clarifications carefully and decided to keep my initial assessment.
> >
> > Sincerely,
> > reviewer

---

### Official Review · Reviewer_vfcG · 2024-11-02

**Soundness:** 3
**Presentation:** 3
**Contribution:** 2
**Rating:** 5
**Confidence:** 4

**Summary:**

The paper introduces a framework that leverages large language models (LLMs) to interpret and generate molecular captions. The work incorporates both 2D and 3D molecular structures to provide a more comprehensive understanding of molecules.

**Strengths:**

1. The paper integrates both 2D and 3D molecular structures to enhance the model's understanding of molecular data.
2. The paper includes detailed figures (Figure 1-3) that clearly explain the method's framework and training scheme.
3. And the analysis of attention maps in Appendix A.4 provides valuable insights into the model's behavior.

**Weaknesses:**

1. Compared to recent related work, such as 3D-MoLM (Li et al., 2024), the innovation in MV-CLAM appears incremental. While the paper claims to incorporate both 2D and 3D molecular structures for a more comprehensive understanding, the approach seems to merely extend the 3D-MoLM framework by introducing 2D components through MAT. The proposed MQ-former architecture does not demonstrate significant structural innovations beyond existing methods. A clearer articulation of the novel contributions and architectural advantages over 3D-MoLM would be necessary to establish the work's originality.
2. The paper considers SMILES as an important molecular modality and notes that "1D SMILES provide compact represen tation of molecular structures", but does not mention SELFIES (Krenn et al., 2020) at all, which has been widely adopted in recent works due to its robust characteristics and tokenization-friendly nature. SELFIES offers inherent robustness and easier tokenization that aligns well with LLMs, making it a potentially more suitable choice for this application.
3. Some images (e.g. the big image at page 18) are not vector graphics and lack titles or captions, which makes it confusing.

**Questions:**

See 'Weaknesses' section.
1. Could the authors provide a more detailed explanation of the novelty of MV-CLAM compared to recent related work?
2. Why was SELFIES not considered as a molecular modality in this work, given its advantages over SMILES in tokenization and alignment with LLMs?

---

> ### Author Response · Authors · 2024-11-24
> **Response to Reviewer vfcG's comments**
>
> **Substantial support on novelty and technical contributions**
> : Please kindly refer to "*General Response (3). Concerning novelty and technical contributions*”.
>
> **Alternative Usage of SELFIES instead of SMILES**
> : Following your recommendation, we conducted additional experiments to compare the performance of SELFIES and SMILES representations within our framework. By replacing SMILES with SELFIES, we trained our model for molecule captioning using identical training hyperparameters on the PubChem324k training dataset. The results demonstrated that models using SMILES outperformed those with SELFIES in terms of BLEU, METEOR, and ROUGE scores. Nonetheless, we appreciate your suggestion and recognize SELFIES as a promising alternative for certain use cases. The results are also available in Appendix A.7.
>
> **Table R7**. Comparison of SELFIES and SMILES as 1D representations
>
> |            | BLEU2 | BLEU4 | METEOR | ROUGE1 | ROUGE2 | ROUGE-L |
> |------------|-------|-------|--------|--------|--------|---------|
> | **SELFIES** | 28.39 | 20.89 | 33.25  | 37.58  | 22.49  | 31.37   |
> | **SMILES**  | **31.75** | **24.48** | **36.54**  | **40.43**  | **25.72**  | **33.79**   |
>
>
> **Clarify images**
> Thank you for pointing out the issues with the images, particularly the one on page 18. We have revised the manuscript to address these concerns. The non-vector graphics have been replaced with vector graphics for improved clarity and quality. Additionally, we have added appropriate titles and captions to all images, including the one on page 18, to enhance their interpretability and provide necessary context.

---

### Official Review · Reviewer_6JAy · 2024-11-03

**Soundness:** 1
**Presentation:** 1
**Contribution:** 2
**Rating:** 3
**Confidence:** 4

**Summary:**

The work proposes a novel multimodal LLM framework MV-CLAM for organic chemistry and MQ-Former — multi-querying transformer model for simultaneous 1D, 2D, and 3D molecular representation learning. Authors show SOTA results in two tasks of molecule-text retrieval and molecule captioning. In addition, authors claim that their approach allows zero-shot molecule editing and molecule-related question answering.

**Strengths:**

New molecular multimodal LLM framework for simultaneous incorporation of 1d 2D and 3D representations.
New Transformer architecture MQ-Former.

**Weaknesses:**

The claim of the state-of-the-art performance for molecule captioning is not satisfied, see the results in [6].
There is no comparison with the other strong retrieval methods for the molecule retrieval task, i.e. RAG.
There are various problems with the Zero-shot editing part of the paper. The task is not formally defined. There are no metrics nor baselines for it.

The QA part is practically absent in the paper, while claimed in the abstract and results parts..
There are many works on molecular conformation generation [1-4], it seems that SMILES and/or 2D-graph representation is enough for neural networks to reconstruct RDKIT conformations almost perfectly. It means that 3D input possibly does not add any new information to the model. There is no comparison of the 1D+2D+3D MQ-Former vs 1D+2D models in the paper.

There is no comparison with other works on multi-modal representation learning for molecules, e.g.: [5].

[1] Zhu, Jinhua, et al. "Direct Molecular Conformation Generation."
[2] Xu, Minkai, et al. "GeoDiff: A Geometric Diffusion Model for Molecular Conformation Generation." International Conference on Learning Representations.
[3] Jing, Bowen, et al. "Torsional diffusion for molecular conformer generation." Advances in Neural Information Processing Systems 35 (2022): 24240-24253.
[4] Lee, Danyeong, et al. "Disco: Diffusion Schrödinger bridge for molecular conformer optimization." Proceedings of the AAAI Conference on Artificial Intelligence. Vol. 38. No. 12. 2024.
[5] Manolache, Andrei, Dragos Tantaru, and Mathias Niepert. "MolMix: A Simple Yet Effective Baseline for Multimodal Molecular Representation Learning." arXiv preprint arXiv:2410.07981 (2024).
[6] Liu, Zhiyuan, et al. "ReactXT: Understanding Molecular" Reaction-ship" via Reaction-Contextualized Molecule-Text Pretraining." arXiv preprint arXiv:2405.14225 (2024).

**Questions:**

1. 3D structures (conformers)

As mentioned in sec. 5.1 you use MMFF for molecular conformation generation.

a. Is it ETKDG geometry generation with further MMFF optimization?
b. Since it is possible to generate several different conformers for a single molecular structure, did you assess the dependence of the model quality on the conformations? Is it necessary to optimize a generated with ETKDG conformer with MMFF?

2.  It would be reasonable to compare your approach for Zero-shot editing with conditional generation models for small molecules.

3. Please, add experiments on the CHEBI-20 benchmark for the captioning task.

---

> ### Author Response · Authors · 2024-11-24
> **Response to Reviewer 6JAy's comments - 1**
>
> **More baseline models, datasets**: Please kindly refer to “Concerning SOTA baseline models” and “Concerning the CheBI20 dataset” in General Response.
>
> **Incorporating multi-modal molecule encoders**: Please kindly refer to “Concerning novelty and technical contributions” in General Response.
>
> **Zero-shot editing task clarification**: We apologize for the lack of explanation in defining the implementation of zero-shot molecule editing. To comprehensively evaluate the quality and robustness of molecular query tokens produced by the MQ-Former module, we defined an auxiliary task focusing on SMILES generation guided by chemical properties. The primary objective of this auxiliary task was to assess whether the specialized language model, after training on molecule captioning, could effectively output valid chemical language (i.e., SMILES) without further tokenization. This evaluation helps determine the utility of the molecular query tokens and their alignment with chemical properties in textual descriptions. Furthermore, by editing molecular descriptions through textual prompts, we aimed to test the model's capacity to adapt its outputs and demonstrate the transferability of its learned molecular representations.
>
> > **Training Phase**: Fine-tune MV-CLAM to print SMILES directly from molecular universal tokens. Conduct the fine-tuning over 4 epochs, using both the PubChem324k pretraining and training dataset.
>
> > **Zero-shot editing evaluation phase**: Provide the model with unseen molecular structures (absent in pretrain/train dataset, provided in dataset described in Appendix) and accompanying chemical prompts (e.g., “The molecule is more soluble in water”). Assess the printed SMILES outputs for chemical property alterations that align with the prompts. (eg. solubility: logP)
>
> For zero-shot editing task clarification, we have now properly included the description on training and evaluation scheme in Section 6.5 and Appendix A.3.
>
> **Comparison with conditional generative models**: We acknowledged that the zero-shot editing task has inherent limitations. Due to the use of LLaMA2’s tokenizer, which is not explicitly trained for SMILES generation, the success rate of generating valid SMILES was not 100%. This underscores the challenges of leveraging a general-purpose tokenizer for such a specialized chemical task. Unlike conditional generative models that are explicitly designed to generate structured outputs like SMILES (e.g., through dedicated tokenizers or architecture adjustments), our approach represents an initial exploration of this capability with a raw, unmodified LLaMA2 architecture. As a result, direct comparisons with conditional generative models are difficult and not entirely fair, as those models are specifically tailored for tasks like molecular generation. Our focus was instead on demonstrating MQ-Former’s ability to bridge molecular and textual representations in a functional and chemically meaningful way. Future work could address these challenges by training custom tokenizers or fine-tuning specific generative models for more robust SMILES generation.
>
> We appreciate the reviewer’s understanding of this distinction and providing an opportunity to highlight the novelty and challenges of our approach.

---

> > ### Author Response · Authors · 2024-11-24
> > **Response to Reviewer 6JAy's comments - 2**
> >
> > **QA results unavailable in main text**: Thank you for your valuable comment. We have evaluated the QA experiment results by comparing our framework, MV-CLAM—integrating the Multi-querying Transformer module with both 3D and 2D molecular encoders—against frameworks that use a single molecular encoder. Specifically, we compared it to:
> >
> > - 3D-MoLM: Q-Former with a 3D molecular encoder
> > - 2D-MoLM: Q-Former with a 2D molecular encoder
> >
> > For each case, including our framework, we reproduced QA results for both non-3D properties (Molecular Weight, LogP, Complexity, and Topological Polar Surface Area) and 3D properties (HOMO, LUMO, HOMO-LUMO gap, and SCF energy), as shown in Tables R5 and R6.
> >
> > **Table R5.** Comparison of QA results for non-3D molecular property prediction. The values in parentheses indicate the validity score.
> > | Model            | Molecular Weight | LogP  | Complexity | Topological Polar Surface Area |
> > |-------------------|-----------------------|------------|------------------|-------------------------------------|
> > | 2D-MoLM          | 47.51 (0.98)         | 0.89 (0.99)| 110.78 (0.99)    | 16.65 (0.99)                        |
> > | 3D-MoLM          | 42.76 (0.96)         | 1.25 (0.96)| 105.03 (0.96)    | 20.97 (0.92)                        |
> > | MQ-Former (Ours) | **21.35 (0.92)**     | **0.69 (0.94)** | **55.14 (0.91)** | **9.65 (0.91)**                     |
> >
> >
> > **Table R6.** Comparison of QA results for 3D molecular property prediction. The values in parentheses indicate the validity score.
> > | Model            | HOMO | LUMO  | HOMO-LUMO | SCF Energy |
> > |-------------------|------------|------------|----------------|-----------------|
> > | 2D-MoLM          | 0.78 (0.99)| 0.47 (0.99)| 0.39 (0.90)    | 0.98 (1.00)     |
> > | 3D-MoLM          | 0.42 (0.99)| 0.44 (0.98)| 1.26 (0.99)    | 1.22 (0.98)     |
> > | MQ-Former (Ours) | **0.35 (0.98)** | **0.42 (0.93)** | **0.35 (0.99)** | **0.32 (0.99)** |
> >
> >
> > Bold text indicates the best performance. As shown in the tables, our framework with MQ-Former achieved the highest scores in both non-3D and 3D molecular property QA tasks by effectively utilizing molecular information from both dimensions. We have now properly updated the result tables as Table 5 in the revised manuscript.

---

> ### Author Response · Authors · 2024-11-24
> **Response to Reviewer 6JAy's comments - 3**
>
> **Question on 3D conformer construction**
> 1. Is it ETKDG geometry generation with further MMFF optimization?
>
> > Yes, the 3D conformers in our study were generated using RDKit’s ETKDG (Extended-Torsion Distance Geometry with additional restraints) method, which incorporates stereochemical rules and experimental torsion-angle preferences for more realistic initial geometries. After embedding the conformer with ETKDG, further refinement was performed using the MMFF force field to optimize bond lengths, angles, and torsional strains, resulting in lower-energy conformations. This process is widely used in prior studies involving 3D molecular modeling and prediction tasks, as it produces chemically meaningful and energetically favorable conformers. Thank you for the opportunity to clarify the 3D conformer construction process. We have now briefly described the process in the PubChem324K section (5.1 Datasets).
>
> 2. Since it is possible to generate several different conformers for a single molecular structure, did you assess the dependence of the model quality on the conformations?
>
> > We acknowledge that a single molecular structure can have multiple valid conformers, each representing a different local energy minimum. In this study, we used one representative conformer per molecule, specifically the lowest-energy conformer obtained after MMFF optimization. While we did not explicitly assess the dependence of model quality on different conformers, this single-conformer approach is consistent with many previous works regarding molecule captioning. Ensuring the use of a chemically plausible, low-energy conformer minimizes variability across datasets. Evaluating the impact of multiple conformers or conformer ensembles on model performance could be a valuable direction for future research.
>
> 3. Is it necessary to optimize a conformer generated with ETKDG using MMFF?
>
> > Although ETKDG generates high-quality 3D conformers with chemically meaningful geometries, additional MMFF optimization is a common practice in prior studies to further refine conformers by minimizing steric clashes and optimizing geometries in terms of potential energy. This step ensures that the conformers better approximate their true physical structures, which can improve the downstream prediction of molecular properties. Therefore, MMFF optimization was considered an essential step in our workflow to align with established best practices and enhance model reliability.

---

### Author Response · Authors · 2024-11-24
**General Response (1). Concerning SOTA baseline models - 1**

We sincerely appreciate the constructive feedback regarding the comparison with state-of-the-art (SOTA) models. **We chose 3D-MoLM [1] and UniMoT [2] as our baseline models because they are the most closely aligned with our model in terms of structural similarity and training data, allowing for the most equitable evaluation regarding our novel cross-modal projector, MQ-Former.** We appreciate the suggestion to implement additional SOTA methods and have carefully considered their inclusion.

While additional comparisons with other SOTA methods remain important, there are differences in dataset composition and preprocessing pipelines across suggested models (Table R1). Even within the PubChem dataset, prior models adopted different preprocessing procedures, leading to variations in the number of molecule entities. As a result, previous works also have demonstrated inconsistencies in the selection of baseline models. **To ensure a fair comparison, we aligned our selection with UniMoT, the most recent model trained on the same dataset.** We updated Table 1 in the revised manuscript - presented as Table R2 - to include MoleculeSTM [3] and MolCA [4]. Similarly, Table 2 in the revised manuscript - shown as Table R3 - was updated to include InstructMol [5] and MolCA as baseline models, following the UniMoT paper.

We recognize the value of retraining other models under the same dataset condition for a rigorous comparison. However, training a single model and performing additional fine-tuning would exceed the available timeline. Therefore, we concentrated our efforts on the most feasible and impactful evaluations within the given discussion timeframe. We prioritized validating the key contributions and novelty of our proposed MQ-Former model through ablation studies and targeted experiments that justify its design and effectiveness. These experiments highlight the distinct advantages of our approach in leveraging 2D and 3D molecular representations to align with textual information as shown in Tables R4-R6, and the updated Appendix Figures 7 and 13 in the revised manuscript.



**Table R1.** Pretraining datasets of baseline models
| Model        | Dataset - #pretrain                         |
|--------------|---------------------------------------------|
| MolCA        | PubChem-MolCA-298k                         |
| 3DMoLM       | PubChem-3DMoLM-301k                        |
| UniMoT       | PubChem-3DMoLM-301k                        |
| MolBind [6]     | PubChem-MolBind-319k                       |
| TextChem T5 [7] | CheBI-Train                                |
| GitMol [8]      | PubChem+CheBI-GitMol-320k → 90k (multimodal)|
| Atomas [9]      | PubChem-Atomas-243k → 51k (high-quality, leak-free) |
| Presto [10]      | PubChem-Presto-326k                        |
| AMORE [11]       | CheBI-Train                                |
| SciFive [12]     | CheBI-Train                                |
| nacho [13]       | Mol-Instructions                           |

---

> ### Author Response · Authors · 2024-11-24
> **General Response (1). Concerning SOTA baseline models - 2**
>
> We provide updated table for molecule-text retrieval and molecule captioning.
>
> **Table R2.** Updated molecule-text retrieval performance.
> | Model            | Retrieval in Batch (M2T) | Retrieval in Batch (T2M) | Retrieval in Test Set (M2T) | Retrieval in Test Set (T2M) |
> |-------------------|--------------------------|---------------------------|-----------------------------|-----------------------------|
> |                  | ACC      | R@20          | ACC      | R@20          | ACC      | R@20          | ACC      | R@20          |
> | **1D SMILES**    |          |               |          |               |          |               |          |               |
> | Sci-BERT         | 85.32    | 98.74         | 84.20    | 98.43         | 41.67    | 87.31         | 40.18    | 86.77         |
> | KV-PLM           | 86.05    | 98.63         | 85.21    | 98.47         | 42.80    | 88.46         | 41.67    | 87.80         |
> | **2D Graph**     |          |               |          |               |          |               |          |               |
> | MoMu-S           | 87.58    | 99.24         | 86.44    | 99.38         | 47.29    | 90.77         | 48.13    | 89.92         |
> | MoMu-K           | 88.23    | 99.41         | 87.29    | 99.42         | 48.47    | 91.64         | 49.46    | 90.73         |
> | MoleculeSTM*     | 90.50    | 99.60         | 88.60    | 99.50         | 52.70    | 92.90         | 53.20    | 92.50         |
> | MolCA*           | 92.60    | 99.80         | 91.30    | 99.50         | 67.90    | 94.40         | 68.60    | 93.30         |
> | **2D Graph + Tokenizer** |   |               |          |               |          |               |          |               |
> | UniMoT           | _93.60_  | **100.0**     | 92.70    | 99.40         | _69.50_  | _96.30_       | 69.80    | 94.40         |
> | **3D Conformer** |          |               |          |               |          |               |          |               |
> | 3D-MoLM          | 93.50    | **100.0**     | _92.89_  | _99.59_       | 69.05    | 95.91         | _70.13_  | _94.88_       |
> | **2D Graph + 3D Conformer** |               |          |               |          |               |          |               |
> | MV-CLAM          | **96.57**| _99.95_       | **97.03**| **99.95**     | **76.32**| **96.57**     | **77.03**| **96.42**     |
>
>
> **Table R3.** Updated molecule captioning performance.
> | Model                    | BLEU-2       | BLEU-4       | ROUGE-1      | ROUGE-2      | ROUGE-L      | METEOR       |
> |--------------------------|--------------|--------------|--------------|--------------|--------------|--------------|
> | **1D SMILES**            |              |              |              |              |              |              |
> | MolT5-Small              | 22.53        | 15.23        | 30.44        | 13.45        | 20.30        | 23.98        |
> | MolT5-Base               | 24.51        | 16.61        | 32.19        | 14.04        | 21.35        | 26.10        |
> | MolT5-Large              | 25.87        | 17.28        | 34.07        | 16.42        | 23.41        | 28.04        |
> | Llama2-7B†               | 27.01        | 20.94        | 35.76        | 20.68        | 28.88        | 32.11        |
> | **2D Graph**             |              |              |              |              |              |              |
> | MoMu-Small               | 22.86        | 16.01        | 30.98        | 13.65        | 20.75        | 24.35        |
> | MoMu-Base                | 24.74        | 16.77        | 32.45        | 14.62        | 22.09        | 27.16        |
> | MoMu-Large               | 26.34        | 18.01        | 34.75        | 16.86        | 24.76        | 28.73        |
> | 2D-MoLM†                 | 27.15        | 21.19        | 36.02        | 20.76        | 29.12        | 32.28        |
> | InstructMol*             | 18.90        | 11.70        | 27.30        | 11.80        | 17.80        | 21.30        |
> | MolCA-Small*             | 25.90        | 17.50        | 34.40        | 16.60        | 23.90        | 28.50        |
> | MolCA-Large*             | 28.60        | 21.30        | 36.20        | 21.40        | 29.70        | 32.60        |
> | **2D Graph + Tokenizer** |              |              |              |              |              |              |
> | UniMoT                   | _31.30_      | _23.80_      | _37.50_      | _23.70_      | _33.60_      | _34.80_      |
> | **3D Conformer**         |              |              |              |              |              |              |
> | 3D-MoLM                  | 30.32        | 22.52        | 36.84        | 22.32        | 31.23        | 33.06        |
> | **2D Graph + 3D Conformer** |           |              |              |              |              |              |
> | MV-CLAM                  | **31.75**    | **24.48**    | **40.43**    | **25.72**    | **33.79**    | **36.54**    |

---

> > ### Comment · Reviewer_6JAy · 2024-11-27
> >
> > Thank you for clarifying some of my concerns.
> > Still, the following remains unclear:
> > 1. Why do the metrics from the UniMolT paper for MolCA models differ so much from the original paper? I suppose relying on an unpublished paper is not a good practice.
> > 2. The same questions for 3D-MoLM models in  the Q&A task. The metrics in the original paper are significantly better
> > 3. The setup of zero-shot editing stays unclear. What are the metrics of this task?

---

> > > ### Author Response · Authors · 2024-11-28
> > > **Response to Reviewer 6JAy's comments 2 - (1/3)**
> > >
> > > We appreciate your thorough review of our rebuttal and the additional insights you have provided. Your feedback has given us an opportunity to further clarify our approach, and we hope our explanations address any remaining concerns effectively.
> > >
> > > > 1.  Why do the metrics from the UniMolT paper for MolCA models differ so much from the original paper? I suppose relying on an unpublished paper is not a good practice.
> > >
> > > As shown in **Table R1**, the performance discrepancies for MolCA compared to the original paper are **entirely expected due to differences in *dataset composition***. Naturally, baseline selections vary across all models because each uses distinct datasets, and the results presented in our paper are based on consistent dataset configurations **following 3D-MoLM**.
> > > While using PubChem as the source, MolCA's preprocessing pipeline results in a *smaller dataset*. Compared to the number of molecules in the PubChem dataset 3D-MoLM, UniMoT, and our model use, MolCA has approximately 3k fewer molecules in the pretraining subset alone. This distinction likely explains why 3D-MoLM, published after MolCA's release, does not include MolCA as a baseline.
> > >
> > > To ensure a fair and comprehensive comparison, we reproduced the results of MolCA under the same dataset setting and obtained similar results to those reported by UniMoT. Notably, UniMoT, which is currently under review for ICLR2025, has provided the most reliable reproduction of MolCA's performance under comparable conditions, and there have been no questions raised about the reliability of its reproduced results in the ongoing review discussions.

---

> > > ### Author Response · Authors · 2024-11-28
> > > **Response to Reviewer 6JAy's comments 2 - (2/3)**
> > >
> > > > 2. The same questions for 3D-MoLM models in the Q&A task. The metrics in the original paper are significantly better.
> > >
> > > We reproduced the Q&A task of 3D-MoLM because, while conducting the Q&A experiments using the code provided by 3D-MoLM, we found that the method used to extract properties (e.g., “The Molecular Weight for the input molecule is 123.18 g/mol”) at the final stage **significantly** affected performance. Specifically, unless we manually extracted the properties one by one, the automated extraction process could lead to errors. However, 3D-MoLM did not provide a clear method for extracting these properties.
> > >
> > > To ensure a fair comparison, we standardized the property extraction process across all models and conducted the performance evaluation using this consistent approach. This allowed us to remove inconsistencies caused by different extraction methods and ensure a more accurate comparison of model performance.
> > >
> > > Although our paper utilizes reproduced results for the above reasons, under the same circumstance using the original (without GPT3.5-enrichment) PubChem324k dataset, our model obtains superior performance compared to 3D-MoLM and 2D-MoLM in the original paper. We provide the comparison of reported official performance for 3D-MoLM, 2D-MoLM and Llama2-7B with ours in TableR10, R11.
> > >
> > > **Table R10.** Comparison of QA results for non-3D molecular property prediction. We report the performance as given in the original paper. We report 3D-MoLM results trained on the original (without GPT3.5 enrichment) dataset.
> > > | Model            | Molecular Weight | LogP  | Complexity | Topological Polar Surface Area |
> > > |-------------------|-----------------------|------------------|------------------|-------------------------------------|
> > > | Llama2-7B          | 22.10 (0.96)         | 1.45 (0.95)| 69.74 (0.93)    | 15.87(0.92)                        |
> > > | 2D-MoLM          | 21.48 (0.94)         | 0.88 (0.96)| 55.74 (0.94)    | 13.52 (0.92)                        |
> > > | 3D-MoLM  (Generalist)        | 19.54 (0.93)        | 0.92 (0.92) | 54.68 (0.90)   | 11.14 (0.92)                  |
> > > | 3D-MoLM  (Specialist)        | **16.18 (0.96)**         | 0.95 (0.96)| **49.15 (0.95)**    | 10.26 (0.94)                        |
> > > | MV-CLAM (Ours) | 21.35 (0.92)     | **0.69 (0.94)** | 55.14 (0.91) | **9.65 (0.91)**                     |
> > >
> > > **Table R11.** Comparison of QA results for 3D molecular property prediction. We report the performance as given in the original paper. We report 3D-MoLM results trained on the original (without GPT3.5 enrichment) dataset.
> > > | Model            | HOMO | LUMO  | HOMO-LUMO | SCF Energy |
> > > |-------------------|------------|------------|----------------|-----------------|
> > > | Llama2-7B         | 1.24 (0.96)| 1.04 (0.95)| 0.88 (0.92)    | 0.70 (0.99)     |
> > > | 2D-MoLM          | 0.92 (0.98)| 0.80 (0.96)| 0.67 (0.93)    | 0.71 (0.99)     |
> > > | 3D-MoLM (Generalist)          | 0.65 (0.94)| 0.41 (0.92)| 0.55 (0.89)    | 0.49 (0.99)     |
> > > | 3D-MoLM (Specialist)          | 0.45 (0.98)| **0.36 (0.96)**| 0.41 (0.94)    | 0.39 (0.99)     |
> > > | MV-CLAM (Ours) | **0.35 (0.98)** | 0.42 (0.93) | **0.35 (0.99)** | **0.32 (0.99)** |
> > >
> > >
> > > Note that we chose to use the original dataset without GPT-enriched descriptions, aligning with observations noted in the 3D-MoLM paper:
> > >
> > > > "The retrieval performance on the PubChem test set appears to be negatively impacted by GPT-3.5 enrichment. We infer that this decline is caused by the enrichment process enlarging the distribution gap between the pretraining and downstream tasks."
> > >
> > > Since our study emphasizes cross-modal alignment over captioning, we concluded that GPT-enriched descriptions might do more harm than good for downstream retrieval performance. This choice ensures our results remain aligned with the study’s objectives and avoid negative impacts on cross-modal tasks.

---

> > > ### Author Response · Authors · 2024-11-28
> > > **Response to Reviewer 6JAy's comments 2 - (3/3)**
> > >
> > > > 3. The setup of zero-shot editing stays unclear. What are the metrics of this task?
> > >
> > > We apologize for any confusion this may have caused.  We would like to clarify our main goal again.
> > >
> > > The primary goal of MV-CLAM is **molecular captioning**  (MolCA, 3d-MoLM, or UniMoT), not molecular generation. We have demonstrated MV-CLAM's effectiveness for molecular captioning through quantitative results.
> > > Additionally, as an *auxiliary case study*, we aim to explore whether the learned model can generalize to molecular generation tasks. This *qualitative experiment* serves as a preliminary investigation, opening to potential future research directions in the area of molecular generation using models designed for captioning. Specifically, we explore how well these tokens align with textual space by directly generating SMILES strings from raw general-purpose LLaMA tokenizers—an approach attempted for the first time in this context.
> > >
> > > The task is evaluated using specific chemical property metrics that correspond to the given instruction prompt. This allows us to qualitatively assess the alignment and efficiency of our generated universal query tokens in guiding meaningful molecular modifications. For instance, in a prompt such as “increase water solubility,” we evaluate the generated molecules based on their logP values, calculated using RDKit. The metric is defined as follows: If the generated molecule has a lower logP value than the original (indicating increased solubility), the result is considered a valid shot.
> > >
> > > The overall task is assessed based on two criteria:
> > >
> > > 1. Validity of the generated SMILES strings: The ability to construct valid molecular structures.
> > >
> > > 2. Success in achieving the desired chemical property change: As measured by property-specific metrics like logP for solubility or other relevant descriptors depending on the prompt.
> > >
> > > These chemical property metrics provide a systematic way to determine whether the generated molecules align with the textual editing instructions, while also showcasing how well the universal query tokens bridge textual and chemical spaces.
> > >
> > > We sincerely value your time in reassessing our revisions. If you have any additional comments or suggestions, we would greatly appreciate your continued engagement during this discussion phase.

---

### Author Response · Authors · 2024-11-24
**General Response (2). Concerning the CheBI20 dataset**

We appreciate the reviewer’s comments and acknowledge the importance of using diverse and extensive datasets for evaluating the generalizability of our model. In this work, we chose to validate our MV-CLAM framework for molecule captioning solely on the PubChem324k dataset, **following the baseline approach (3D-MoLM) we built upon**, which also reports captioning performances on the PubChem324k dataset.

Our decision to exclude the ChEBI-20 dataset was primarily motivated by the following considerations:
1. Data Redundancy and Leakage Concerns
> ChEBI-20 is derived from PubChem324k dataset, with additional manual curation for specific biological contexts. Since ChEBI-20 is essentially a subset of PubChem, there is an inherent overlap between the two datasets. This overlap raises potential concerns about data redundancy and leakage when training and evaluating on these datasets together.

2. Evaluation of Molecular Nomenclature
> Unlike the PubChem324k dataset, which retains molecular names and provides a broader variety of molecular structures, ChEBI-20 replaces molecular names with generic placeholders such as “the molecule.” While this emphasizes molecular properties, it limits the evaluation of the model’s ability to connect structural features with accurate molecular nomenclature. Names often encode critical structural information (e.g., functional groups, stereochemistry, or ring systems), making them an essential aspect of evaluating a model’s understanding of molecular structures.

Taking these considerations into account, we believe that using the **PubChem dataset provides a rigorous and comprehensive evaluation of our framework’s capabilities in text retrieval, molecule captioning, and downstream tasks**. We have included a detailed discussion in **Appendix A.2** on our dataset selection process and the rationale for excluding ChEBI20. We emphasize that *validating our captioning performance on the PubChem dataset alone is sufficient, as it offers a significantly larger and more diverse set of molecular descriptions, enabling a robust assessment of the generalizability and efficacy of our model*. We deeply appreciate the reviewers for thoughtful suggestions.


References
> [1] Li et al., "Towards 3D Molecule-Text Interpretation in Language Models". [2] Zhang et al., "UniMoT: Unified Molecule-Text Language Model with Discrete Token Representation" [3] Liu et al., "Multi-modal Molecule Structure-text Model for Text-based Retrieval and Editing" [4] Liu et al., "MolCA: Molecular Graph-Language Modeling with Cross-Modal Projector and Uni-Modal Adapter". [5] Cao et al., "InstructMol: Multi-Modal Integration for Building a Versatile and Reliable Molecular Assistant in Drug Discovery". [6] Xiao et al., "MolBind: Multimodal Alignment of Language, Molecules, and Proteins" [7] Christofidellis et al., "Unifying Molecular and Textual Representations via Multi-task Language Modelling" [8] Liu et al., "GIT-Mol: A Multi-modal Large Language Model for Molecular Science with Graph, Image, and Text" [9] Zhang et al., "Atomas: Hierarchical Alignment on Molecule-Text for Unified Molecule Understanding and Generation" [10] Cao et al., "PRESTO: Progressive Pretraining Enhances Synthetic Chemistry Outcomes" [11] Ganeeva et al., "Lost in Translation: Chemical Language Models and the Misunderstanding of Molecule Structures" [12] Phan et al., "SciFive: a text-to-text transformer model for biomedical literature" [13] Livne et al., "nach0: Multimodal Natural and Chemical Languages Foundation Model"

---

### Author Response · Authors · 2024-11-24
**General Response (3). Concerning novelty & technical contributions**

We sincerely thank the reviewers for their valuable feedback regarding the novelty and technical contributions of MQ-Former, and we apologize for not clearly conveying these aspects in our initial submission. MQ-Former *aligns molecular and text spaces by sharing self-attention layers between each multi-view structural information (2D and 3D) and text*, generating a universal query token interpretable to LLMs. The **simultaneous yet separate alignment of 2D and 3D structural representation to text**, is only applicable with our novel MQ-Former, and ensures a balanced incorporation of  abundant structural information among the differing views. This design addresses key challenges in multi-modal learning to *align multiple modalities with minimal information loss*. We have demonstrated this through our case study of captions, attention visualization and ablation study of using single view molecular embeddings mentioned in Section 6.3.

To further address your comments and demonstrate the uniqueness of our approach, we have conducted two additional studies: 1)  Embedding space visualization to demonstrate MQ-Former preserves modality-specific information in accordance to textual semantics. 2) Comparison ablation that utilizes multi-view molecular embeddings (2D+3D embeddings concatenated) in the former Q-Former framework. This necessitates the usage of employing MQ-Former with an additional branch, differentiating it from Q-Former.

1. **Embedding Space Visualization**

> We analyzed the embeddings of the 2D, 3D queries, and our universal query tokens in the latent space alongside the corresponding word embeddings from textual descriptions. Specifically, we focused on highly 2D- or 3D-related words from the textual captions described in Appendix Figure 6-7 (Case Study 1. Attention visualization).
For 2D-related words, the distance between the embeddings followed the trend: 2D < 2D+3D < 3D.Conversely, for 3D-related words, the trend was: 3D < 2D+3D < 2D.These observations confirm that MQ-Former successfully preserves modality-specific information while aligning it with textual semantics, highlighting the interplay between 2D and 3D molecular views in multi-modal learning. We have included the visualization results in **Appendix Figure 7**.

2. **Comparison with Multi-View Representations in Q-Former Framework**

> To highlight the necessity of MQ-Former, we conducted an ablation study comparing our architecture with a variant that aligns multi-view molecular representations using a single Q-Former module. The suggested MolMix [14] (a multimodal representation learning framework) by reviewer 6JAy lacks pretrained weights for molecular embeddings. Hence, we leveraged the 2D embeddings from MAT and the 3D embeddings from Uni-Mol, concatenating these representations before projecting them into the textual space using the Q-Former. We combined the results with the former ablation conducted in Table 3.

> Overall, the study emphasizes that while single-view embeddings (e.g., 2D or 3D alone) capture important molecular information, they lack the comprehensive representation needed for captioning tasks requiring multi-faceted insights. Moreover, unlike the concatenation-based approach, MQ-Former preserves the rich, distinct representations of molecular views. That is, *the simultaneous alignment of these embeddings in the shared textual space enhances the preservation of intricate molecular properties*.  This design facilitates more fine-grained alignment with text, maintaining diversified information, which results in higher-quality captions across all evaluated metrics (Table R4). Overall, MQ-Former enables the preservation of detailed and diverse molecular representations, facilitating precise alignment with textual descriptions and delivering superior performance across captioning task.

**Table R4**. Comparison of molecule captioning performance: multi-view embeddings aligned with Q-Former
|                        | BLEU2  | BLEU4  | METEOR | ROUGE1 | ROUGE2 | ROUGE-L |
|----------------------------|--------|--------|--------|--------|--------|--------|
| **2D + Q-Former** | 29.72 | 22.26 | 34.22 | 38.22 | 23.45 |  31.61  |
| **3D+ Q-Former** | 29.45 | 22.03 | 33.79 | 37.86 |  23.11 | 31.83  |
| **Multi-view (2D&3D) + Q-Former** | 29.80  | 22.70  | 35.49  | 39.07  | 24.92  | 33.09  |
| **MV-CLAM**            | **31.75**  | **24.48**  | **36.54**  | **40.43**  | **25.72**  | **33.79**  |

This novelty ensures more precise molecule-text understanding, distinguishing *MQ-Former as a robust and effective solution for molecular captioning*. We appreciate the opportunity to clarify our contributions and have incorporated these findings into the revised manuscript organized in **Appendix A.4. Effectiveness of MQ-Former**.


Reference
> [14] Manolache, Andrei, Dragos Tantaru, and Mathias Niepert. "MolMix: A Simple Yet Effective Baseline for Multimodal Molecular Representation Learning."

---

### Meta-Review · Area_Chair_ykyg · 2024-12-14

**Metareview:**

This paper proposes a novel multimodal LLM framework, MV-CLAM, for organic chemistry, and MQ-Former, a multi-query transformer model for simultaneous 1D, 2D, and 3D molecular representation learning, aiming to provide a more comprehensive understanding of molecules.

The idea of integrating 1D, 2D, and 3D molecular information is interesting. The design of MQ-Former involves adapting the Q-Former technique from multimodal learning. However, the experimental results are not convincing enough to demonstrate the superiority of the proposed method, due to issues such as missing baselines, insufficient ablation studies, and reliance on a single dataset. Additionally, the technical novelty is somewhat limited. Therefore, I do not recommend the acceptance of this paper.

**Additional Comments On Reviewer Discussion:**

During the rebuttal period, reviewer expressed a lack of enthusiasm for the paper. AC tried to engage the discussion but `Reviewer 6JAy, vfcG, R1Xc` didn't make any response.

AC reviewed the paper, along with the comments and responses, to make the final decision.

---

### Decision · Program_Chairs · 2025-01-22

Reject